# Efficient Degradation-agnostic Image Restoration via Channel-Wise Functional Decomposition and Manifold Regularization

**Bin Ren**[1,2]   **Yawei Li**[3]   **Xu Zheng**[4]   **Yuqian Fu**[5]   **Danda Pani Paudel**[5]   **Hong Liu**[6]*
**Ming-Hsuan Yang**[7]   **Luc Van Gool**[5]   **Nicu Sebe**[2]
[1]Mohamed bin Zayed University of Artificial Intelligence   [2]University of Trento   [3]ETH Zürich
[4]HKUST (GZ)   [5]INSAIT, Sofia University "St. Kliment Ohridski"
[6]Peking University   [7] University of California, Merced

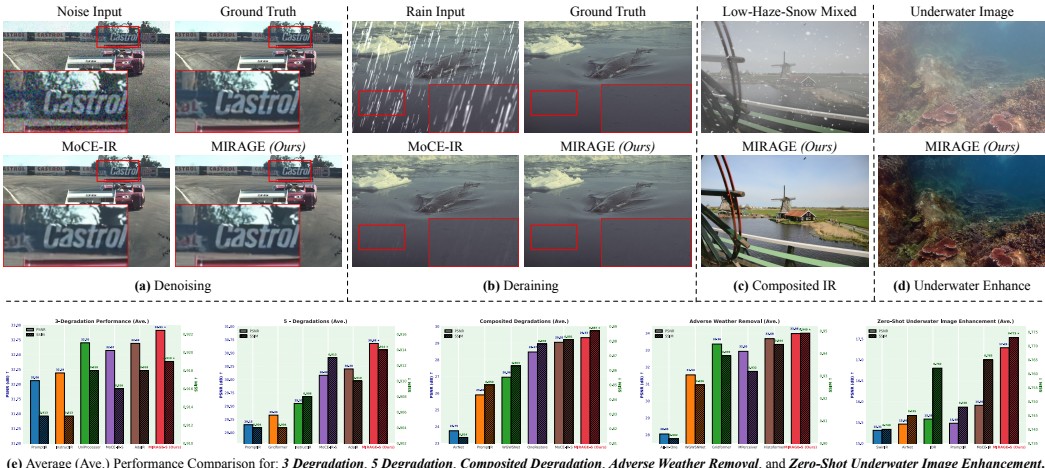

Figure 1: *(a)-(d)*: Visual comparison for Denoising, Deraining, Composited Degradations (low-light, haze, and snow), and underwater image enhancement. *(e)*: The average PSNR and SSIM comparison across 4 challenging all-in-one and 1 zero-shot settings (Please zoom in for a better view).

## Abstract

Degradation-agnostic image restoration aims to handle diverse corruptions with one unified model, but faces fundamental challenges in balancing efficiency and performance across different degradation types. Existing approaches either sacrifice efficiency for versatility or fail to capture the distinct representational requirements of various degradations. We present MIRAGE, an efficient framework that addresses these challenges through two key innovations. First, we propose a channel-wise functional decomposition that systematically repurposes channel redundancy in attention mechanisms by assigning CNN, attention, and MLP branches to handle local textures, global context, and channel statistics, respectively. This principled decomposition enables degradation-agnostic learning while achieving superior efficiency-performance trade-offs. Second, we introduce manifold regularization that performs cross-layer contrastive alignment in Symmetric Positive Definite (SPD) space, which empirically improves feature consistency and generalization across degradation types. Extensive experiments demonstrate that MIRAGE achieves state-of-the-art performance with remarkable efficiency, outperforming existing methods in various all-in-one IR settings while offering a scalable and generalizable solution for challenging unseen IR scenarios.

---

* indicates corresponding author: Hong Liu <hongliu@pku.edu.cn>.

# 1 INTRODUCTION

Image Restoration (IR) aims to recover clean images from inputs degraded by diverse real-world corruptions such as noise, blur, haze, rain, and low-light conditions (Zamir et al., 2022; Li et al., 2023a; Ren et al., 2024; Potlapalli et al., 2024). A central challenge is *degradation-agnostic restoration*: developing a single model that can generalize across heterogeneous degradations. Despite recent progress, existing approaches often face an efficiency–performance dilemma. On the one hand, heavyweight designs based on prompts, instructions, or large vision–language models provide versatility but incur substantial computational cost (Potlapalli et al., 2024; Zamfir et al., 2025; Jiang et al., 2025). On the other hand, lightweight solutions improve efficiency at the expense of restoration quality (Li et al., 2022; Tang et al., 2025b). Achieving both robustness and efficiency within a unified framework remains an open problem.

This difficulty can be better understood from two complementary perspectives. First, different degradation types impose fundamentally different representational requirements: additive corruptions (*e.g.*, noise, rain) primarily affect local textures, multiplicative distortions (*e.g.*, haze, low-light) require global context modeling, and kernel-based degradations (*e.g.*, blur) call for multi-scale structural reasoning. At the same time, basic architectural modules exhibit distinct inductive biases: convolutional filters excel at local texture modeling, attention mechanisms capture long-range dependencies, and MLPs enhance channel statistics. This motivates the insight that *an effective restoration model should systematically align distinct modules with complementary representational functions*. Second, recent studies reveal substantial redundancy in attention-based models, particularly along the channel dimension (Venkataramanan et al., 2024; Dong et al., 2021). Many channels encode overlapping information, suggesting that this redundancy could be *repurposed* rather than discarded. Leveraging this observation allows for architectures that remain compact while preserving expressive capacity. These observations highlight that unified IR benefits not only from adding new modules, but from a principled reorganization of existing capacity based on redundancy patterns and complementary inductive biases. This perspective motivates our design philosophy in MIRAGE, where representational roles are explicitly aligned with structural evidence rather than heuristic module stacking.

Building on these insights, we present MIRAGE, an efficient framework for degradation-agnostic image restoration. MIRAGE introduces two components. (i) *Channel-wise functional decomposition*, where the input feature map is partitioned along the channel dimension and processed by three specialized branches: convolution for local textures, attention for global context, and MLP for channel statistics. This structured decomposition repurposes redundant capacity into complementary roles, yielding both interpretability and strong efficiency–performance trade-offs. (ii) *Manifold regularization*, a cross-layer contrastive strategy that leverages natural feature pairs within the model. Inspired by deeply supervised networks (Lee et al., 2015), we hypothesize that natural contrastive pairs exist between shallow and latent representations. Shallow features preserve fine spatial details but are sensitive to noise, while latent features are more abstract and semantically stable; aligning them encourages more robust shared representations. Importantly, rather than computing contrastive loss in Euclidean space, which may distort similarity when comparing structured representations, we operate in the Symmetric Positive Definite (SPD) manifold space. This formulation provides a more faithful alignment of representations, leading to improved generalization across degradation types. Overall, MIRAGE provides a structurally grounded view of unified IR, where representational capacity is allocated and aligned based on statistical evidence at both the spatial and depth levels.

Extensive experiments across five degradation settings show that MIRAGE achieves state-of-the-art performance with remarkable efficiency: our model has only 6M parameters, more than five times smaller than recent prompt-based baselines, while also generalizing well to unseen scenarios such as underwater image enhancement. Both the visual and per-setting PSNR results are shown in Fig. 1.

Our contributions are summarized as follows:

- We propose a principled channel-wise functional decomposition strategy that aligns convolution, attention, and MLP with distinct representational roles, enabling efficient and effective degradation-agnostic restoration.

- We introduce manifold regularization through cross-layer contrastive alignment between shallow and latent features. We exploit natural contrastive pairs within the model, and per-

form this alignment in the SPD manifold space rather than Euclidean space, providing more faithful representation similarity and improved generalization across diverse degradations.

- We conduct comprehensive experiments across single, mixed, and unseen degradation settings, establishing MIRAGE as a strong and practical baseline for all-in-one IR.

## 2 RELATED WORK

**Image Restoration with Various Architectures.** IR addresses the ill-posed problem of retoring high-quality images from degraded inputs and has long been a core task in computer vision with broad applications (Richardson, 1972; Banham & Katsaggelos, 1997; Xie et al., 2025; Li et al., 2023b; Zamfir et al., 2024). Early methods relied on model-based formulations with handcrafted priors, but deep learning has shifted the field toward data-driven approaches, including regression-based (Lim et al., 2017; Lai et al., 2017; Liang et al., 2021; Chen et al., 2021; Li et al., 2023a; Zhang et al., 2024) and generative pipelines (Gao et al., 2023; Wang et al., 2023b; Luo et al., 2023; Yue et al., 2023; Zhao et al., 2024). These methods employ diverse backbones: convolutional networks for local structures (Dong et al., 2015; Zhang et al., 2017b;a; Wang et al., 2018), MLPs and state space models for channel or sequential dependencies (Tu et al., 2022; Guo et al., 2024a; Zhu et al., 2024; Gu & Dao, 2023; Dao & Gu, 2024; Tang et al., 2025a), and Transformers for long-range interactions (Liang et al., 2021; Ren et al., 2023; Li et al., 2023a; Zamir et al., 2022; Dosovitskiy et al., 2020; Liu et al., 2023; Shi et al., 2025), achieving promising results. Despite these advances, most IR solutions remain degradation-specific, addressing tasks such as denoising (Zhang et al., 2019), dehazing (Wu et al., 2021), deraining (Jiang et al., 2020), or deblurring (Kong et al., 2023), motivating the need for unified frameworks that generalize across diverse degradations while remaining efficient.

**Degradation-agnostic Image Restoration.** While training task-specific models for individual degradations can be effective, it is impractical to maintain separate models for each corruption. Real-world images often suffer from mixed degradations, making independent treatment infeasible, and task-specific approaches further increase computational and storage costs, amplifying their environmental footprint. To overcome these limitations, the emerging field of degradation-agnostic IR focuses on single-blind models capable of handling multiple degradation types simultaneously (Zamfir et al., 2025; Zeng et al., 2025; Zheng et al., 2024b; Ren et al., 2026). For example, AirNet (Li et al., 2022) achieves blind All-in-One image restoration by using contrastive learning to derive degradation representations from corrupted images, which are then leveraged to reconstruct clean images. Building on this, IDR (Zhang et al., 2023) tackles the problem by decomposing degradations into fundamental physical components and applying a two-stage meta-learning strategy. More recently, the extra learnable prompt-based paradigm (Potlapalli et al., 2024; Wang et al., 2023a; Li et al., 2023c; Tian et al., 2025) has introduced a visual prompt learning module, enabling a single model to better handle diverse degradation types by leveraging the discriminative capacity of learned visual prompts. Extending this idea, some works further model prompts from a frequency perspective (Cui et al., 2025) or propose more complex architectures with additional datasets (Dudhane et al., 2024). However, visual prompt modules often result in increased training time and decreased efficiency (Cui et al., 2025). Meanwhile, inspired by recent advances in self-supervised learning, several works (Wu et al., 2021; Chen et al., 2022c) have explored contrastive objectives to enhance low-level representations, though mainly within single-task IR scenarios. For the degradation-agnostic setting (Jiang et al., 2025; Li et al., 2022; Chen et al., 2025b; Zhang et al., 2025), the most recent DA-RCOT (Tang et al., 2025b) introduces a contrastive loss applied to residual feature space, illustrating that contrastive signals can also benefit unified IR models. In contrast, our work aims to improve the model's ability to capture representative degradation cues within the SPD space without relying on heavy or complex prompt designs. Our goal in this work is to develop a degradation-agnostic image restorer that remains both computationally efficient and environmentally sustainable.

## 3 PRELIMINARY: DEGRADATION-AWARE ARCHITECTURES FOR IR

**Image Degradation and Restoration.** Image restoration seeks to recover a clean image $\mathbf{x}$ from a degraded observation $\mathbf{y}$:

$$\mathbf{y} = \mathcal{D}(\mathbf{x}) + \mathbf{n}, \tag{1}$$

where $\mathcal{D}(\cdot)$ denotes a degradation operator and $\mathbf{n}$ noise. Real-world degradations are diverse—additive (*e.g.*, Gaussian noise, rain: $\mathbf{y} = \mathbf{x} + \mathbf{n}$), multiplicative (*e.g.*, haze, speckle: $\mathbf{y} = \mathbf{x} \cdot \mathbf{m}$), or convolutional

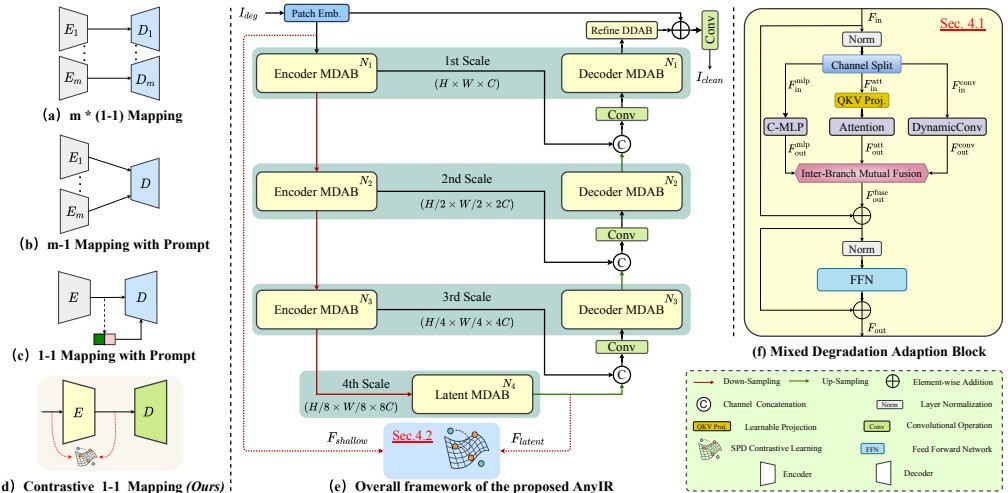

Figure 2: *(a)-(c)*: The most adopted all-in-one image restoration encoder-decoder pipelines. *(d)*: The toy illustration of our SPD contrastive pipeline. *(e)*: The overall framework of the proposed MIRAGE : *i.e.*, a convolutional patch embedding, a U-shape encoder-decoder main body, an extra refined block, and the proposed SPD contrastive learning algorithm. *(f)*: Structure of each mixed degradation adaptation block (MDAB).

*(e.g.*, blur, super-resolution: $\mathbf{y} = \mathbf{k} * \mathbf{x} + \mathbf{n}$) (He et al., 2025). These factors often co-occur and are spatially variant (Zhai et al., 2023), forming compound pipelines:

$$\mathbf{y} = \mathcal{D}_3\big(\mathcal{D}_2(\mathcal{D}_1(\mathbf{x}))\big) + \mathbf{n}. \tag{2}$$

Such complexity demands models that preserve local details while reasoning about global structures.

**Architectural Biases for Degradation Modeling.** Deep networks embody distinct inductive biases: *CNNs* capture local spatial patterns: $\mathbf{y}p = \sum i \in \mathcal{N}(p)w_i \cdot \mathbf{x}_i$, effective for uniform or spatially invariant degradations. *Transformers* exploit global self-attention: $\mathbf{y}i = \sum_j \alpha ij \cdot \mathbf{V}_j$, well-suited for non-uniform, structured degradations (*e.g.* haze, patterned noise). *MLPs*, especially token-mixing forms, apply flexible position-wise mappings: $\mathbf{y} = \mathbf{W}_2 \cdot \phi(\mathbf{W}_1 \cdot \mathbf{x})$, though with weak spatial priors.

Each paradigm shows strengths yet clear limitations—CNNs excel in local fidelity, Transformers in global reasoning, and MLPs in flexible feature interactions, but lack inductive structure. Alone, they are insufficient for complex degradations and often parameter-heavy. Their complementarity motivates unified, degradation-aware architectures that leverage all three for robust IR in the wild.

## 4 THE PROPOSED MIRAGE

The design of MIRAGE is guided by two empirical observations. *(i)* Attention features consistently exhibit low-rank channel redundancy across scales (Fig. 3), indicating that a non-trivial portion of the representational capacity can be reassigned without loss of expressiveness. *(ii)* Different degradations favor complementary inductive biases, *i.e.*, local texture sensitivity, global contextual aggregation, and channel-statistical modulation. These observations motivate a principled partition of feature channels into convolutional, attention, and MLP pathways, allowing each subspace to specialize in the bias it is best suited for while maintaining overall model compactness. In parallel, the depth-asymmetric covariance structures of shallow and latent representations provide a natural basis for cross-layer alignment, for which the SPD formulation offers a geometry-preserving representation.

Prior works either train a separate model per degradation (Fig. 2a), adopt multi-encoder–single-decoder designs that inflate parameters (Fig. 2b), or rely on large-scale prompt-based models with visual/textual cues (Fig. 2c). In contrast, we propose a simple yet effective mixed-backbone architecture (Fig. 2d), which already forms a strong restoration baseline (Sec. 4.1) and is further enhanced by cross-layer contrastive learning in SPD space between shallow and latent features (Sec. 4.2).

### 4.1 MIXED DEGRADATION ADAPTATION BLOCK FOR DEGRADATION-AGNOSTIC IR

---

**Algorithm 1** Mixed Parallel Degradation Adaptation

---

**Require:** $F_{\text{in}}^{\text{att}}$, $F_{\text{in}}^{\text{conv}}$, $F_{\text{in}}^{\text{mlp}}$          $\triangleright$ Input features from three branches
**Ensure:** $F_{\text{out}}$          $\triangleright$ Final fused output
     **[Att] Attention Path**
1: $Q, K, V \leftarrow \text{Linear}(F_{\text{in}}^{\text{att}})$          $\triangleright$ Projection to attention tokens
2: $F_{\text{out}}^{\text{att}} \leftarrow \texttt{Softmax}(\frac{QK^\top}{\sqrt{d}})V$          $\triangleright$ Multi-head self-attention
     **[Conv] Dynamic Convolution Path**
3: $F' \leftarrow \text{Conv1x1}(\text{Norm}(F_{\text{in}}^{\text{conv}}))$          $\triangleright$ Normalization and expansion
4: $\gamma, \beta, \alpha \leftarrow \text{Split}(F')$          $\triangleright$ Gating, intermediate, convolutional paths
5: $\alpha' \leftarrow \text{DynamicDepthwiseConv}(\alpha)$          $\triangleright$ Content-adaptive depthwise conv
6: $\hat{F} \leftarrow \sigma(\gamma/\tau) \cdot \text{Concat}(\beta, \alpha')$          $\triangleright$ Gated local enhancement
7: $F_{\text{out}}^{\text{conv}} \leftarrow \text{Conv1x1}(\hat{F}) + F_{\text{in}}^{\text{conv}}$          $\triangleright$ Residual projection
     **[MLP] MLP Path**
8: $F_{\text{out}}^{\text{mlp}} \leftarrow \text{MLP}(F_{\text{in}}^{\text{mlp}})$          $\triangleright$ Channel-wise transformation brings more non-linearity
     **[Fusion] Inter-Branch Mutual Fusion**
9: $F_{\text{out}}^{\text{att}'} \leftarrow F_{\text{out}}^{\text{att}} + \lambda_{\text{att}} \cdot \sigma(F_{\text{out}}^{\text{conv}} + F_{\text{out}}^{\text{mlp}})$          $\triangleright$ Fuse conv and MLP into attention
10: $F_{\text{out}}^{\text{conv}'} \leftarrow F_{\text{out}}^{\text{conv}} + \lambda_{\text{conv}} \cdot \sigma(F_{\text{out}}^{\text{att}} + F_{\text{out}}^{\text{mlp}})$          $\triangleright$ Fuse attention and MLP into conv
11: $F_{\text{out}}^{\text{mlp}'} \leftarrow F_{\text{out}}^{\text{mlp}} + \lambda_{\text{mlp}} \cdot \sigma(F_{\text{out}}^{\text{att}} + F_{\text{out}}^{\text{conv}})$          $\triangleright$ Fuse attention and conv into MLP
     **Output Projection**
12: $F_{\text{out}}^{\text{fuse}} \leftarrow \text{Project}(\text{Concat}(F_{\text{out}}^{\text{att}'}, F_{\text{out}}^{\text{conv}'}, F_{\text{out}}^{\text{mlp}'}))$          $\triangleright$ Final unified representation
13: **return** $F_{\text{out}}^{\text{fuse}}$

---

**Redundancy in MHAs Opens Opportunities for Hybrid Architectures.** Redundancy has long been recognized as a fundamental limitation in multi-head self-attention (MHA), the core building block of Transformers, in both NLP and vision domains (Nguyen et al., 2022b;a; Xiao et al., 2024; Brödermann et al., 2025; Wang et al., 2022; Venkataramanan et al., 2024). Prior studies indicated that not all attention heads contribute equally, *i.e.*, some are specialized and crucial, while others can be pruned with negligible impact. *This inherently implies redundancy in the channel dimension, as MHA outputs are concatenated along this axis.* To empirically verify this redundancy in the context of IR, we analyze intermedi-

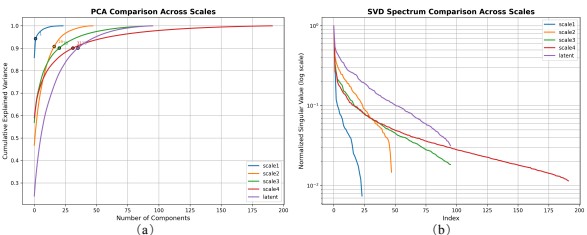

Figure 3: Channel redundancy analysis across multiple feature scales. (a) Cumulative explained variance curves from PCA applied to the channel dimension of features from 1-4 scales and one latent scale. (b) Normalized singular value spectra (in log scale) of the same features via SVD. Latent feature in both plots means the channel-wise projected 4th Scale feature.

ate features from a lightweight attention-only model (details in the Appendix A). Specifically, we compute cumulative explained variance via PCA and normalized singular value spectra via SVD across multiple feature scales. Fig. 3(a) shows earlier scales (*e.g.*, 1st Scale) need far fewer principal components to retain most variance, suggesting high redundancy. Fig. 3(b) further supports this, with a sharper singular value decay at shallower stages, indicating stronger low-rank structure in channel-wise representations. Even at the deepest stage (*e.g.*, 4th Scale), achieving 90% variance requires only 31 of 192 components ($\approx 16\%$), confirming redundancy persists throughout.

This insight motivates a departure from traditional head/channel pruning. Instead of discarding redundant capacity, we propose to *repurpose* it by splitting the channel dimension into three parts and feeding them into distinct architectural branches, *i.e.*, attention, convolution, and MLP. This hybrid formulation leverages complementary inductive biases and makes full use of available representational space, offering a principled and efficient alternative to the previous pure MSA-based designs.

**Parallel Design Brings More Efficiency.** As shown in Lines 1–8 of Alg. 1, we instantiate this idea through a structurally parallel design that simultaneously exploits complementary inductive biases. As illustrated in Fig. 2(f), the input feature $F_{\text{in}} \in \mathbb{R}^{h \times w \times c}$ is evenly partitioned along the channel dimension into three sub-tensors (*i.e.*, $F_{\text{in}}^{\text{att}}$, $F_{\text{in}}^{\text{conv}}$, and $F_{\text{in}}^{\text{mlp}}$,), which are then processed in parallel by

Figure 4: *(a)-(d)*: The channel-wise similarity matrix from the 1st Scale ($H \times W \times C$) to the 4th Scale ($H/8 \times W/8 \times 8C$). *(e)*: The channel-wise similarity matrix of (d) after channel-wise projection.

attention, convolution, and MLP branches. Each branch operates only on its allocated fraction of channels, substantially reducing computational cost, while its architectural heterogeneity enriches the representational space. This parallel decomposition achieves a favorable balance between efficiency and expressiveness, in contrast to prior designs that rely on purely attention-based processing.

**Inter-Branch Mutual Fusion Injects Expressivity Before FFN.** While the parallel design improves efficiency and modularity, it reduces interaction across branches. To mitigate this, Lines 9–13 of Alg. 1 introduce an inter-branch fusion mechanism, where each branch is enhanced via gated aggregation of the rest, modulated by learnable coefficients $\lambda$. This introduces cross-path context blending, reinforcing feature complementarity before unification, forming an effective pre-FFN encoder.

Compared to the attention-only models, the fused output in Alg. 1 introduces richer interactions. This enhances the model's ability to fit complex degradation mappings, making it more suitable for mixed or ambiguous degradations. Subsequently, layer normalization, a feed-forward network (FFN), and a residual connection are applied: $F_{\text{out}} = \text{FFN}(\text{Norm}(F_{\text{out}}^{\text{fuse}})) + F_{\text{out}}^{\text{fuse}}$. This sequence stabilizes feature distributions and further boosts expressiveness.

## 4.2 SHALLOW-LATENT CONTRASTIVE LEARNING VIA SPD MANIFOLD ALIGNMENT

The unified IR model requires a single backbone to process degradations that depend on fundamentally different representational levels. Shallow layers primarily encode degradation-specific, fine-grained structures, whereas deeper layers become more semantic and statistically stable. This inherent depth asymmetry introduces representation drift when multiple degradations share the same feature space, motivating a mechanism that explicitly enforces cross-stage consistency. We therefore treat shallow and latent features as complementary views of the underlying signal and align them to stabilize the shared representation space, thereby improving generalization across heterogeneous degradations.

**Shallow-Latent Feature Pairs are Naturally Contrastive Pairs.** Features extracted at different depths exhibit fundamentally different statistical properties. As shown in Fig. 4, shallow-stage features (*e.g.*, Scale1) present *sparse and decorrelated channel distributions*, while deeper layers (*e.g.*, Scale4) become *increasingly redundant and concentrated*. This trend is quantitatively supported by the effective rank ratio across scales, which increases from only 4.2% ($1/24$ at 1st Scale) to 16.1% ($31/192$ at 4th Scale). However, by compressing the deep features through a lightweight MLP, we obtain a latent representation with a notably higher rank ratio of 36.5% ($35/96$), indicating a more decorrelated and expressive embedding. This structural disparity between sparse, localized shallow features and compressed, semantic latent ones naturally defines a contrastive pairing without requiring augmentation. We leverage this depth-asymmetric contrast to impose consistency across stages, enabling better semantic alignment and stronger representational generalization under complex degradation conditions. *Note that this study is conducted under noise degradation; however, similar trends are consistently observed for other degradations as well*. See the appendix for more details.

**SPD Manifold Space Contrastive Learning Leads to More Discriminative Representations.** To enhance representation consistency across depth, we introduce a contrastive objective defined over SPD (Symmetric Positive Definite) manifold features. We note that the goal here is not to perform full Riemannian optimization along SPD geodesics. Instead, we adopt a lightweight formulation that retains the key second-order structure of covariance matrices while keeping training stable and efficient. Strict geodesic contrastive learning typically requires repeated log/exp mappings and matrix decompositions, which incur considerable overhead in large low-level vision models. Our approach strikes a practical balance by preserving essential SPD structure before projection. Specifically, in our work, given shallow features $F_{\text{shallow}} \in \mathbb{R}^{C_s \times H \times W}$ and latent features $F_{\text{latent}} \in \mathbb{R}^{C_l \times H' \times W'}$, we first reduce their channel dimensions via $1 \times 1$ convolutions. The resulting tensors are reshaped into

feature matrices $X_s, X_l \in \mathbb{R}^{C \times N}$ with $N = H \times W$, and their second-order statistics are computed as:

$$\mathbf{C}_s = \frac{1}{N-1}(X_s - \mu_s)(X_s - \mu_s)^\top + \epsilon\mathbf{I}, \quad \mathbf{C}_l = \frac{1}{N'-1}(X_l - \mu_l)(X_l - \mu_l)^\top + \epsilon\mathbf{I}, \quad (3)$$

where $\mu$ is the mean across spatial dimensions, and $\epsilon\mathbf{I}$ ensures numerical stability and positive definiteness. The SPD matrices $\mathbf{C}_s, \mathbf{C}_l \in \mathbb{R}^{C \times C}$ are vectorized and projected to a contrastive embedding space via shallow 1-layer MLPs:

$$z_s = \text{Norm}(W_s \cdot \text{vec}(\mathbf{C}_s)), \quad z_l = \text{Norm}(W_l \cdot \text{vec}(\mathbf{C}_l)), \quad (4)$$

where $W_s, W_l$ are learnable projection layers, and $\text{Norm}(\cdot)$ denotes $\ell_2$-normalization. We then apply an InfoNCE-style contrastive loss to align the shallow and latent embeddings:

$$\mathcal{L}_{\text{SPD}} = -\log \frac{\exp\left(\text{sim}(z_s, z_l)/\tau\right)}{\sum_{z_l'} \exp\left(\text{sim}(z_s, z_l')/\tau\right)}, \quad (5)$$

where $\text{sim}(\cdot, \cdot)$ denotes cosine similarity and $\tau$ a temperature parameter. Unlike Euclidean contrastive learning, which views features as flat vectors, our SPD-based method preserves second-order channel dependencies, providing richer structural supervision. This regularization aligns local and semantic features across depth, enhances discriminability, and *introduces no additional inference cost*.

## 5 EXPERIMENTS

We conduct experiments adhering to the protocols of prior general image restoration works (Potlapalli et al., 2024; Zhang et al., 2023) under 5 settings: *(a) 3 Degradations), (b) 5 Degradations), (c) Mixed Degradation, (d) Adverse Weather Removal*, and *(e) Zero-Shot*. The implementation and experimental details, and the dataset description are provided in the appendix. Our code, checkpoints, and visual results are available via: https://github.com/Amazingren/MIRAGE.

### 5.1 SOTA COMPARISON.

**3 Degradations.** We evaluate our method against others listed in Tab. 1, all trained on three degradations: dehazing, deraining, and denoising. MIRAGE consistently outperforms all the comparison methods, even for those with the assistance of language, multi-task, or prompts. Notably, even our **6M** tiny model outperforms our baseline PromptIR by **0.71dB** on average. Our 10M small model achieves the best performance across all the metrics, with **60%** fewer parameters compared MoCE-IR. Compared to DA-RCOT (Tang et al., 2025b), which performs contrastive learning over residual feature space, MIRAGE achieves consistently better restoration quality while using substantially fewer parameters (10M vs. 50M). This highlights the efficiency and effectiveness of our SPD-based cross-layer alignment despite its more compact design.

**5 Degradations.** Extending the 3 tasks with deblurring and low-light enhancement (Li et al., 2022; Zhang et al., 2023), we evaluate our MIRAGE 's performance in a more challenging 5-degradation setting. Tab. 2 shows that MIRAGE -S surpasses PromptIR (Potlapalli et al., 2024), MoCE-IR-S (Zamfir et al., 2025), AdaIR (Cui et al., 2025), and VLU-Net (Zeng et al., 2025) by **1.53dB**, **0.6dB**, **0.48dB**, and **0.57dB** on average, with fewer parameters. Our tiny model (6M) also achieves a second-best average PSNR against MoCE-IR (25M) and surpasses all other methods, including those aided by additional modalities, multi-task learning, or pretraining.

**Mixed Degradations.** To better approximate real-world conditions, we extend OneRestore (Guo et al., 2024b) to cover both diverse single degradations (rain, haze, snow, low light) and composite cases with multiple degradations per image, yielding eleven distinct restoration settings. As shown in Tab. 3, MIRAGE consistently outperforms leading approaches including AirNet (Li et al., 2022), PromptIR (Potlapalli et al., 2024), WGWSNet (Zhu et al., 2023), WeatherDiff (Özdenizci & Legenstein, 2023), OneRestore (Guo et al., 2024b), and MoCE-IR (Zamfir et al., 2025). Specifically, our Tiny (6M) and Small (10M) models outperform OneRestore (Guo et al., 2024b) (6M) by **0.39 dB** and **0.86dB** on average. Compared to the recent SOTA MoCE-IR (Zamfir et al., 2025) (11M), our Small model achieves **0.28dB** higher performance with fewer parameters (10M vs. 11M). These results highlight the effectiveness of our method, particularly for complex mixed degradations.

Table 1: *Comparison to state-of-the-art on three degradations.* PSNR (dB, ↑) and SSIM (↑) metrics are reported on the full RGB images. **Best** performances is highlighted. '-' means unreported results.

| Method | Venue. | Params. | Dehazing SOTS | | Deraining Rain100L | | Denoising BSD68$_{\sigma=15}$ | | BSD68$_{\sigma=25}$ | | BSD68$_{\sigma=50}$ | | Average | |
|---|---|---|---|---|---|---|---|---|---|---|---|---|---|---|
| BRDNet (Tian et al., 2020) | NN'20 | - | 23.23 | .895 | 27.42 | .895 | 32.26 | .898 | 29.76 | .836 | 26.34 | .693 | 27.80 | .843 |
| LPNet (Gao et al., 2019) | CVPR'19 | - | 20.84 | .828 | 24.88 | .784 | 26.47 | .778 | 24.77 | .748 | 21.26 | .552 | 23.64 | .738 |
| FDGAN (Dong et al., 2020) | AAAI'20 | - | 24.71 | .929 | 29.89 | .933 | 30.25 | .910 | 28.81 | .868 | 26.43 | .776 | 28.02 | .883 |
| DL (Fan et al., 2019) | TPAMI'19 | 2M | 26.92 | .931 | 32.62 | .931 | 33.05 | .914 | 30.41 | .861 | 26.90 | .740 | 29.98 | .876 |
| MPRNet (Zamir et al., 2021) | CVPR'21 | 16M | 25.28 | .955 | 33.57 | .954 | 33.54 | .927 | 30.89 | .880 | 27.56 | .779 | 30.17 | .899 |
| AirNet (Li et al., 2022) | CVPR'22 | 9M | 27.94 | .962 | 34.90 | .967 | 33.92 | .933 | 31.26 | .888 | 28.00 | .797 | 31.20 | .910 |
| NDR (Yao et al., 2024) | TIP'24 | 28M | 25.01 | .860 | 28.62 | .848 | 28.72 | .826 | 27.88 | .798 | 26.18 | .720 | 25.01 | .810 |
| PromptIR (Potlapalli et al., 2024) | NeurIPS'23 | 36M | 30.58 | .974 | 36.37 | .972 | 33.98 | .933 | 31.31 | .888 | 28.06 | .799 | 32.06 | .913 |
| MoCE-IR-S (Zamfir et al., 2025) | CVPR'25 | 11M | 30.98 | .979 | 38.22 | .983 | 34.08 | .933 | 31.42 | .888 | 28.16 | .798 | 32.57 | .916 |
| AdaIR (Cui et al., 2025) | ICLR'25 | 29M | 31.06 | .980 | 38.64 | .983 | 34.12 | .935 | 31.45 | .892 | 28.19 | .802 | 32.69 | .918 |
| MoCE-IR (Zamfir et al., 2025) | CVPR'25 | 25M | 31.34 | .979 | 38.57 | .984 | 34.11 | .932 | 31.45 | .888 | 28.18 | .800 | 32.73 | .917 |
| DA-RCOT (Tang et al., 2025b) | TPAMI'25 | 50M | 31.26 | .977 | 38.36 | .983 | 33.98 | .934 | 31.33 | .890 | 28.10 | .801 | 32.60 | .917 |
| MIRAGE -T (*Ours*) | ICLR'26 | 6M | 31.81 | **.982** | 38.44 | .983 | 34.05 | .935 | 31.40 | **.892** | 28.14 | .802 | 32.77 | .919 |
| MIRAGE -S (*Ours*) | ICLR'26 | 10M | **31.86** | .981 | **38.94** | **.985** | **34.12** | **.935** | **31.46** | .891 | **28.19** | **.803** | **32.91** | **.919** |
| Methods with the assistance of vision language, multi-task learning, natural language prompts, or multi-modal control | | | | | | | | | | | | | | |
| DA-CLIP (Luo et al., 2024) | ICLR'24 | 125M | 29.46 | .963 | 36.28 | .968 | 30.02 | .821 | 24.86 | .585 | 22.29 | .476 | - | - |
| Art$_{PromptIR}$ (Wu et al., 2024) | ACM MM'24 | 36M | 30.83 | .979 | 37.94 | .982 | 34.06 | .934 | 31.42 | .891 | 28.14 | .801 | 32.49 | .917 |
| InstructIR-3D (Conde et al., 2024) | ECCV'24 | 16M | 30.22 | .959 | 37.98 | .978 | 34.15 | .933 | 31.52 | .890 | 28.30 | .804 | 32.43 | .913 |
| UniProcessor (Duan et al., 2025) | ECCV'24 | 1002M | 31.66 | .979 | 38.17 | .982 | 34.08 | .935 | 31.42 | .891 | 28.17 | .803 | 32.70 | .918 |
| VLU-Net (Zeng et al., 2025) | CVPR'25 | 35M | 30.71 | .980 | 38.93 | .984 | 34.13 | .935 | 31.48 | .892 | 28.23 | .804 | 32.70 | .919 |
| RamIR (Tang et al., 2025a) | Applied'25 | 21.7M | 31.29 | .977 | 38.16 | .981 | 34.04 | .931 | 31.61 | .891 | 28.19 | .801 | 32.65 | .916 |

Table 2: *Comparison to state-of-the-art on five degradations.* PSNR (dB, ↑) and SSIM (↑) metrics are reported on the full RGB images with (*) denoting general image restorers, others are specialized all-in-one approaches. **Best** performance is highlighted.

| Method | Venue | Params. | Dehazing SOTS | | Deraining Rain100L | | Denoising BSD68$_{\sigma=25}$ | | Deblurring GoPro | | Low-Light LOLv1 | | Average | |
|---|---|---|---|---|---|---|---|---|---|---|---|---|---|---|
| NAFNet* (Chen et al., 2022a) | ECCV'22 | 17M | 25.23 | .939 | 35.56 | .967 | 31.02 | .883 | 26.53 | .808 | 20.49 | .809 | 27.76 | .881 |
| DGUNet* (Mou et al., 2022) | CVPR'22 | 17M | 24.78 | .940 | 36.62 | .971 | 31.10 | .883 | 27.25 | .837 | 21.87 | .823 | 28.32 | .891 |
| SwinIR* (Liang et al., 2021) | ICCVW'21 | 1M | 21.50 | .891 | 30.78 | .923 | 30.59 | .868 | 24.52 | .773 | 17.81 | .723 | 25.04 | .835 |
| Restormer* (Zamir et al., 2022) | CVPR'22 | 26M | 24.09 | .927 | 34.81 | .962 | 31.49 | .884 | 27.22 | .829 | 20.41 | .806 | 27.60 | .881 |
| MambaIR* (Guo et al., 2024a) | ECCV'24 | 27M | 25.81 | .944 | 36.55 | .971 | 31.41 | .884 | 28.61 | .875 | 22.49 | .832 | 28.97 | .901 |
| DL (Fan et al., 2019) | TPAMI'19 | 2M | 20.54 | .826 | 21.96 | .762 | 23.09 | .745 | 19.86 | .672 | 19.83 | .712 | 21.05 | .743 |
| Transweather | CVPR'22 | 38M | 21.32 | .885 | 29.43 | .905 | 29.00 | .841 | 25.12 | .757 | 21.21 | .792 | 25.22 | .836 |
| TAPE (Liu et al., 2022) | ECCV'22 | 1M | 22.16 | .861 | 29.67 | .904 | 30.18 | .855 | 24.47 | .763 | 18.97 | .621 | 25.09 | .801 |
| AirNet (Li et al., 2022) | CVPR'22 | 9M | 21.04 | .884 | 32.98 | .951 | 30.91 | .882 | 24.35 | .781 | 18.18 | .735 | 25.49 | .847 |
| IDR (Zhang et al., 2023) | CVPR'23 | 15M | 25.24 | .943 | 35.63 | .965 | 31.60 | .887 | 27.87 | .846 | 21.34 | .826 | 28.34 | .893 |
| PromptIR (Potlapalli et al., 2024) | NeurIPS'23 | 36M | 30.41 | .972 | 36.17 | .970 | 31.20 | .885 | 27.93 | .851 | 22.89 | .829 | 29.72 | .901 |
| MoCE-IR-S (Zamfir et al., 2025) | CVPR'25 | 11M | 31.33 | .978 | 37.21 | .978 | 31.25 | .884 | 28.90 | .877 | 21.68 | .851 | 30.08 | .913 |
| AdaIR (Cui et al., 2025) | ICLR'25 | 29 | 30.53 | .978 | 38.02 | .981 | 31.35 | .889 | 28.12 | .858 | 23.00 | .845 | 30.20 | .910 |
| MoCE-IR (Zamfir et al., 2025) | CVPR'25 | 25M | 30.48 | .974 | 38.04 | .982 | 31.34 | .887 | **30.05** | **.899** | 23.00 | .852 | 30.58 | **.919** |
| DA-RCOT (Tang et al., 2025b) | TPAMI'25 | 50M | 30.96 | .975 | 37.87 | .980 | 31.23 | .888 | 28.68 | .872 | 23.25 | .836 | 30.40 | .911 |
| MIRAGE -T (*Ours*) | ICLR'26 | 6M | 31.35 | .979 | 38.24 | .983 | 31.35 | .891 | 27.98 | .850 | 23.11 | .854 | 30.41 | .912 |
| MIRAGE -S (*Ours*) | ICLR'26 | 10M | **31.45** | **.980** | **38.92** | **.985** | **31.41** | **.892** | 28.10 | .858 | **23.59** | **.858** | **30.68** | .914 |
| Methods with the assistance of natural language prompts or multi-task learning | | | | | | | | | | | | | | |
| InstructIR-5D (Conde et al., 2024) | ECCV'24 | 16M | 36.84 | .973 | 27.10 | .956 | 31.40 | .887 | 29.40 | .886 | 23.00 | .836 | 29.55 | .908 |
| Art$_{PromptIR}$ (Wu et al., 2024) | ACM MM'24 | 36M | 29.93 | .908 | 22.09 | .891 | 29.43 | .843 | 25.61 | .776 | 21.99 | .811 | 25.81 | .846 |
| VLU-Net (Zeng et al., 2025) | CVPR'25 | 35M | 30.84 | .980 | 38.54 | .982 | 31.43 | .891 | 27.46 | .840 | 22.29 | .833 | 30.11 | .905 |
| RamIR (Tang et al., 2025a) | Applied'25 | 21.7M | 31.09 | .978 | 37.56 | .979 | 31.44 | .886 | 28.82 | .878 | 22.02 | .828 | 30.18 | .910 |

**Adverse Weather Removal.** Following (Valanarasu et al., 2022; Zhu et al., 2023), We test our MIRAGE on three challenging deweathering tasks: snow removal, rain streak and fog removal, and raindrop removal. Tab. 4 shows the comparison of our MIRAGE and other state-of-the-art methods. MIRAGE consistently outperforms existing methods across almost all datasets except PSNR for RainDrop. The performance gains over multiple weather degradations demonstrate the effectiveness of MIRAGE in handling diverse weather conditions. Especially, **0.30dB** improvement on PSNR over Histoformer (Sun et al., 2024) and **1.05dB** improvements over MPerceiver (Ai et al., 2024).

**Zero-Shot Setting.** We evaluate our method's generalization under a challenging zero-shot setting with real-world underwater images. As shown in Tab. 5, MIRAGE -S achieves 17.29 dB and 0.773 SSIM, surpassing MoCE-IR (Zamfir et al., 2025) by **+1.38dB** PSNR, while being more compact.

Table 3: *Comparison to state-of-the-art on composited degradations.* PSNR (dB, ↑) and SSIM (↑) are reported on the full RGB images. Our method consistently outperforms even larger models, with favorable results in composited degradation scenarios.

| Method | Params. | CDD11-Single | | | | CDD11-Double | | | | | CDD11-Triple | | Avg. |
|---|---|---|---|---|---|---|---|---|---|---|---|---|---|
| | | Low (L) | Haze (H) | Rain (R) | Snow (S) | L+H | L+R | L+S | H+R | H+S | L+H+R | L+H+S | |
| AirNet | 9M | 24.83 .778 | 24.21 .951 | 26.55 .891 | 26.79 .919 | 23.23 .779 | 22.82 .710 | 23.29 .723 | 22.21 .868 | 23.29 .901 | 21.80 .708 | 22.24 .725 | 23.75 .814 |
| PromptIR | 36M | 26.32 .805 | 26.10 .969 | 31.56 .946 | 31.53 .960 | 24.49 .789 | 25.05 .771 | 24.51 .761 | 24.54 .924 | 23.70 .925 | 23.74 .752 | 23.33 .747 | 25.90 .850 |
| WGWSNet | 26M | 24.39 .774 | 27.90 .982 | 33.15 .964 | 34.43 .973 | 24.27 .800 | 25.06 .772 | 24.60 .765 | 27.23 .955 | 27.65 .960 | 23.90 .772 | 23.97 .771 | 26.96 .863 |
| WeatherDiff | 83M | 23.58 .763 | 21.99 .904 | 24.85 .885 | 24.80 .888 | 21.83 .756 | 22.69 .730 | 22.12 .707 | 21.25 .868 | 21.99 .868 | 21.23 .716 | 21.04 .698 | 22.49 .799 |
| OneRestore | 6M | 26.48 .826 | 32.52 .990 | 33.40 .964 | 34.31 .973 | 25.79 .822 | 25.58 .799 | 25.19 .789 | 29.99 .957 | 30.21 .964 | 24.78 .788 | 24.90 .791 | 28.47 .878 |
| MoCE-IR | 11M | 27.26 .824 | 32.66 .990 | 34.31 .970 | 35.91 .980 | 26.24 .817 | 26.25 .800 | 26.04 .793 | 29.93 .964 | 30.19 .970 | 25.41 .789 | 25.39 .790 | 29.05 .881 |
| MIRAGE (ours) | 6M | 27.13 .830 | 32.39 .989 | 34.23 .969 | 35.57 .978 | 26.04 .823 | 26.21 .807 | 26.07 .799 | 29.49 .962 | 29.72 .967 | 25.17 .793 | 25.41 .793 | 28.86 .883 |
| MIRAGE (ours) | 10M | **27.41 .833** | **33.12 .992** | **34.66 .971** | **35.98 .981** | **26.55 .828** | **26.53 .810** | **26.33 .803** | **30.32 .965** | **30.27 .969** | **25.59 .801** | **25.86 .799** | **29.33 .887** |

Table 4: Comparisons for *4-task adverse weather removal*. Missing values are denoted by '–'.

| Method | Venue | Snow100K-S | | Snow100K-L | | Outdoor-Rain | | RainDrop | | Average | |
|---|---|---|---|---|---|---|---|---|---|---|---|
| | | PSNR | SSIM | PSNR | SSIM | PSNR | SSIM | PSNR | SSIM | PSNR | SSIM |
| All-in-One (Li et al., 2020) | CVPR'20 | – | – | 28.33 | .882 | 24.71 | .898 | 31.12 | .927 | 28.05 | .902 |
| TransWeather (Valanarasu et al., 2022) | CVPR'22 | 32.51 | .934 | 29.31 | .888 | 28.83 | .900 | 30.17 | .916 | 30.20 | .909 |
| Chen *et al.* (Chen et al., 2022b) | CVPR'22 | 34.42 | .947 | 30.22 | .907 | 29.27 | .915 | 31.81 | .931 | 31.43 | .925 |
| WGWSNet (Zhu et al., 2023) | CVPR'23 | 34.31 | .946 | 30.16 | .901 | 29.32 | .921 | 32.38 | .938 | 31.54 | .926 |
| WeatherDiff$_{64}$ (Özdenizci & Legenstein, 2023) | TPAMI'23 | 35.83 | .957 | 30.09 | .904 | 29.64 | .931 | 30.71 | .931 | 31.57 | .931 |
| WeatherDiff$_{128}$ (Özdenizci & Legenstein, 2023) | TPAMI'23 | 35.02 | .952 | 29.58 | .894 | 29.72 | .922 | 29.66 | .923 | 31.00 | .923 |
| AWRCP (Ye et al., 2023) | ICCV'23 | 36.92 | .965 | 31.92 | .934 | 31.39 | .933 | 31.93 | .931 | 33.04 | .941 |
| GridFormer (Wang et al., 2024) | IJCV'24 | 37.46 | .964 | 31.71 | .923 | 31.87 | .933 | 32.39 | .936 | 33.36 | .939 |
| MPerceiver (Ai et al., 2024) | CVPR'24 | 36.23 | .957 | 31.02 | .916 | 31.25 | .925 | **33.21** | .929 | 32.93 | .932 |
| DTPM (Ye et al., 2024) | CVPR'24 | 37.01 | .966 | 30.92 | .917 | 30.99 | .934 | 32.72 | .944 | 32.91 | .940 |
| Histoformer (Sun et al., 2024) | ECCV'24 | 37.41 | .966 | 32.16 | .926 | 32.08 | .939 | 33.06 | .944 | 33.68 | .944 |
| MIRAGE -S *(Ours)* | ICLR'26 | **37.97** | **.973** | **32.33** | **.929** | **32.82** | **.949** | 32.78 | **.945** | **33.98** | **.949** |

Table 5: *Zero-Shot* Cross-Domain Underwater Image Enhancement Results.

| Method | PSNR (↑) | SSIM (↑) |
|---|---|---|
| SwinIR (Liang et al., 2021) | 15.31 | .740 |
| NAFNet (Chu et al., 2022) | 15.42 | .744 |
| Restormer (Zamir et al., 2022) | 15.46 | .745 |
| AirNet (Li et al., 2022) | 15.46 | .745 |
| IDR (Zhang et al., 2023) | 15.58 | .762 |
| PromptIR (Potlapalli et al., 2024) | 15.48 | .748 |
| MoCE-IR (Zamfir et al., 2025) | 15.91 | .765 |
| MIRAGE -S *(Ours)* | **17.29** | **.773** |

Table 6: *Complexity Analysis.* FLOPs are computed on an image of size $224 \times 224$ using a NVIDIA Tesla A100 (40G) GPU.

| Method | PSNR (↑) | Memory (↓) | Params. (↓) | FLOPs (↓) |
|---|---|---|---|---|
| AirNet (Li et al., 2022) | 31.20 | 4829M | 8.93M | 238G |
| PromptIR (Potlapalli et al., 2024) | 32.06 | 9830M | 35.59M | 132G |
| IDR (Zhang et al., 2023) | - | 4905M | 15.34M | 98G |
| AdaIR (Cui et al., 2025) | - | 9740M | 28.79M | 124G |
| MoCE-IR-S (Zamfir et al., 2025) | 32.51 | 4263M | 11.48M | 37G |
| MoCE-IR (Zamfir et al., 2025) | 32.73 | 6654M | 25.35M | 75G |
| MIRAGE -T *(Ours)* | 32.77 | **3729M** | **6.21M** | **16G** |
| MIRAGE -S *(Ours)* | **32.91** | 4810M | 9.68M | 27G |

Importantly, our model never sees underwater data during training, yet our adaptive modeling not only fits mixed degradations but also transfers robustly to unseen conditions. Besides, we also followed the same experimental setting introduced by UniRestore Chen et al. (2025a) for the generalization ability evaluation. Meanwhile, the real-world evaluation presented in Tab. D shows that MIRAGE generalizes reliably to real-world, camera-captured degradations.

**Efficiency Comparison.** Tab. 6 compares PSNR, memory, parameters, and FLOPs. Our Tiny model (MIRAGE -T), with only 6.21M parameters and 16G FLOPs, delivers the best efficiency–performance trade-off, outperforming all prior methods, including larger models like PromptIR (Potlapalli et al., 2024) and MoCE-IR-S (Zamfir et al., 2025). It surpasses MoCE-IR-S by **+0.26 dB** while using less than half the computation, and even our Small variant (MIRAGE -S) exceeds full MoCE-IR in both PSNR (**+0.18dB**) and FLOPs (27G vs. 75G). These results confirm that our design achieves strong restoration quality without compromising efficiency.

**Visual Comparison.** MIRAGE effectively restores fine structural details and reliably suppresses subtle visual artifacts across diverse and unseen degradations (Fig. 1 and appendix).

## 5.2 ABLATION ANALYSIS & DISCUSSION

**Components ablation.** Tab. 7 shows starting from an attention-only setting (32.23 dB, 19.89M), we progressively integrate each module while reducing complexity.

Removing the dynamic convolution branch (*w/o DynamicConv*) causes a 0.56 dB drop, indicating its importance for local spatial modeling. The channel-wise MLP (*w/o C-MLP*) also plays a critical role, with a 0.38 dB performance loss. Naive concatenation (*w/o Fusion*) leads to a further 0.20 dB drop, confirming that explicit feature integration is more effective. On the regularization side, removing contrastive learning (*w/o CL & SPD*) or replacing SPD with Euclidean alignment degrades performance

Table 7: *Ablation Study* of MIRAGE -T under the 3-Degradation Setting with Tiny model.

| Ablaton | Params. | Results | |
|---|---|---|---|
| | | PSNR (dB, ↑) | SSIM(↓) |
| att-only *(Ours)* | 19.89 M | 32.23 (-0.54) | .912 |
| *w/o* DynamicConv | 9.43 M | 32.21 (-0.56) | .911 |
| *w/o* C-MLP | 7.01 M | 32.39 (-0.38) | .913 |
| *w/o* Fusion (*i.e.* Cat()-Only) | 5.71 M | 32.57 (-0.20) | .914 |
| *w/o* CL & SPD | 5.80M | 32.63 (-0.14) | .916 |
| *w/o* SPD (CL Euclidean) | 6.10M | 32.53 (-0.24) | .914 |
| MIRAGE -T *(Full)* | 6.21M | 32.77 | .919 |

by 0.14 dB and 0.24 dB, indicating that structure-agnostic contrastive learning can misguide optimization, while manifold-aware alignment provides consistent benefits. Overall, each component contributes to the final performance. Our full model offers the best balance between accuracy and efficiency with only 6.21M parameters and 32.77 dB PSNR.

**Why shallow–latent Contrastive Alignment Matters.** Different degradations rely on different feature levels: denoising and deraining benefit from shallow, texture-rich features, while dehazing and low-light enhancement require deeper semantic features; deblurring needs both. This heterogeneity makes unified modeling challenging. We therefore introduce contrastive alignment between shallow and latent stages to encourage semantic coordination. When shallow features dominate (*e.g.*, denoising), alignment guides latent features to be more task-relevant; when latent features dominate (*e.g.*, dehazing), shallow features

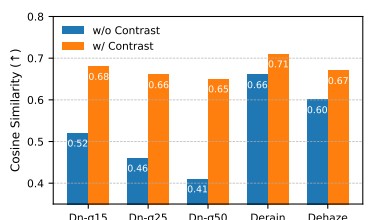

Figure 5: Shallow–latent cosine similarity across degradations. Contrastive alignment improves feature correlation.

inherit semantic consistency (Bertasius et al., 2015). Fig. 5 validates that contrastive alignment improves shallow–latent correlation, validating its necessity for cross-degradation generalization.

**Why Euclidean Fails and Why SPD Works? (Deraining Case Study)** Euclidean contrastive learning collapses shallow–latent alignment by enforcing indiscriminate similarity, reducing both diagonal and off-diagonal terms to trivial constants, and erasing task cues. SPD, by aligning covariance matrices on a Riemannian manifold, preserves second-order dependencies and guides updates along meaningful directions. In the deraining case (Figure 6), Euclidean CL degenerates into near-constant similarity (off-diag 0.00237, ratio 0.99), while SPD maintains diagonal dominance and non-trivial off-diagonal structure (0.0787, ratio 0.149), producing coherent patterns.

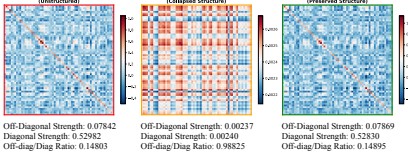

Figure 6: Shallow–latent similarity under three settings: (a) w/o CL (unstructured; off-diag 0.0784, ratio 0.148), (b) Euclidean CL (collapsed; off-diag ≈ 0.0024, ratio 0.99), (c) SPD CL (preserved; off-diag 0.0787, ratio 0.149).

## 6   CONCLUSION

We presented MIRAGE , an efficient framework for degradation-agnostic image restoration that achieves a favorable balance between robustness and efficiency. Through channel-wise functional decomposition, the model repurposes redundant capacity into convolution-, attention-, and MLP-based branches, enabling complementary modeling of local textures, global context, and channel-wise statistics. To further enhance cross-degradation generalization, we introduced manifold regularization, aligning shallow and latent features in the SPD manifold space for more consistent and discriminative representations. Extensive experiments across diverse degradations, including mixed and unseen scenarios, demonstrate that MIRAGE achieves state-of-the-art performance. Inspired by the metaphor of a mirage, *i.e.*, revealing the hidden reality beneath visual distortions, our framework learns degradation-agnostic representations by balancing global, local, and channel-wise information, providing a scalable foundation for future research in degradation-agnostic IR.

ACKNOWLEDGMENTS

This work was partially supported by the FIS project GUIDANCE (Debugging Computer Vision Models via Controlled Cross-modal Generation) (No. FIS2023-03251).

ETHICS STATEMENT

Our work focuses on general-purpose image restoration, aiming to improve efficiency and robustness across diverse degradation types. The intended positive impact includes deployment in low-resource or safety-critical scenarios such as mobile photography, remote sensing, medical imaging, and environmental monitoring. At the same time, we recognize that improved restoration techniques could be misused for deceptive content editing or large-scale surveillance. We encourage responsible use of our method and provide our models and code with appropriate licenses and documentation to support transparency and ethical adoption. No personally identifiable or sensitive data were used in this research.

REPRODUCIBILITY STATEMENT

We aim to ensure reproducibility and transparency of our results. The MIRAGE framework is implemented in PyTorch with standard training protocols and evaluation metrics. Detailed descriptions of the architecture, training settings, datasets, and baselines are provided in the main paper and supplementary material. Upon acceptance, we will release the full code, pretrained models, and instructions for reproducing all reported results, including ablation studies and comparisons. Random seeds and hardware details are also documented to facilitate faithful replication.

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

## A    EXPERIMENTAL PROTOCOLS

### A.1    DATASETS

**3 Degradation Datasets.** For both the All-in-One and single-task settings, we follow the evaluation protocols established in prior works Li et al. (2022); Potlapalli et al. (2024); Zamfir et al. (2025), utilizing the following datasets: For image denoising in the single-task setting, we combine the BSD400 Arbelaez et al. (2010) and WED Ma et al. (2016) datasets, and corrupt the images with Gaussian noise at levels $\sigma \in \{15, 25, 50\}$. BSD400 contains 400 training images, while WED includes 4,744 images. We evaluate the denoising performance on BSD68 Martin et al. (2001) and Urban100 Huang et al. (2015). For single-task deraining, we use Rain100L Yang et al. (2020), which provides 200 clean/rainy image pairs for training and 100 pairs for testing. For single-task dehazing, we adopt the SOTS dataset Li et al. (2018), consisting of 72,135 training images and 500 testing images. Under the All-in-One setting, we train a unified model on the combined set of the aforementioned training datasets for 130 epochs and directly test it across all three restoration tasks.

**5 Degradation Datasets.** The 5-degradation setting is built upon the 3-degradation setting, with two additional tasks included: deblurring and low-light enhancement. For deblurring, we adopt the GoPro dataset Nah et al. (2017), which contains 2,103 training images and 1,111 testing images. For low-light enhancement, we use the LOL-v1 dataset Wei et al. (2018), consisting of 485 training images and 15 testing images. Note that for the denoising task under the 5-degradation setting, we report results using Gaussian noise with $\sigma = 25$. The training takes 150 epochs.

**Composited Degradation Datasets.** Regarding the composite degradation setting, we use the CDD11 dataset Guo et al. (2024b). CDD11 consists of 1,183 training images for: *(i) 4 kinds of single-degradation types:* haze (H), low-light (L), rain (R), and snow (S); *(ii) 5 kinds of double-degradation types:* low-light + haze (l+h), low-light+rain (L+R), low-light + snow (L+S), haze + rain (H+R), and haze + snow (H+S). *(iii) 2 kinds of Triple-degradation type:* low-light + haze + rain (L+H+R), and low-light + haze + snow (L+H+S). We train our method for 170 epochs (fewer than 200 epochs than MoCE-IR Zamfir et al. (2025)), and we keep all other settings unchanged.

**Adverse Weather Removal Datasets.** For the deweathering tasks, we follow the experimental setups used in TransWeather Valanarasu et al. (2022) and WGWSNet Zhu et al. (2023), evaluating the performance of our approach on multiple synthetic datasets. We assess the capability of MIRAGE across three challenging tasks: snow removal, rain streak and fog removal, and raindrop removal. The training set, referred to as "AllWeather", is composed of images from the Snow100K Liu et al. (2018), Raindrop Qian et al. (2018), and Outdoor-Rain Li et al. (2019b) datasets. For testing, we evaluate our model on the following subsets: Snow100K-S (16,611 images), Snow100K-L (16,801

Table A: The details our the tiny and small version of our MIRAGE . FLOPs are computed on an image of size $224 \times 224$ using a NVIDIA Tesla A100 (40G) GPU.

|  | MIRAGE -T | MIRAGE -S |
|---|---|---|
| The Number of the MDAB crosses 4 scales | [3, 5, 5, 7] | [3, 5, 5, 7] |
| The Input Embedding Dimension | 24 | 30 |
| The FFN Expansion Factor | 2 | 2 |
| The Number of the Refinement Blocks | 2 | 3 |
| Params. ($\downarrow$) | 6.21M | 9.68 M |
| FLOPs ($\downarrow$) | 16 G | 27 G |

images), Outdoor-Rain (750 images), and Raindrop (249 images). Same as Histoformer Sun et al. (2024), we train MIRAGE on "AllWeather" with 300,000 iterations.

**Zero-Shot Underwater Image Enhancement Dataset.** For the zero-shot underwater image enhancement setting, we follow the evaluation protocol of DCPT JiaKui et al. (2025) by directly applying our model, trained under the 5-degradation setting, on the UIEB dataset Li et al. (2019a) without any finetuning. UIEB consists of two subsets: 890 raw underwater images with corresponding high-quality reference images, and 60 challenging underwater images. We evaluate our zero-shot performance on the 890-image subset with available reference images.

## A.2 IMPLEMENTATION DETAILS

**Implementation Details.** Our MIRAGE framework is designed to be end-to-end trainable, removing the need for multi-stage optimization of individual components. The architecture adopts a robust 4-level encoder-decoder structure, with a varying number of Mixed Degradation Attention Blocks (MDAB) at each level—specifically $[3, 5, 5, 7]$ from highest to lowest resolution in the Tiny variant. Following prior works Potlapalli et al. (2024); Zamfir et al. (2025), we train the model for 120 epochs with a batch size of 32 in both the 3-Degradation All-in-One and single-task settings. The optimization uses a combination of $L_1$ and Fourier loss, optimized with Adam Kingma & Ba (2015) (initial learning rate of $2 \times 10^{-4}$, $\beta_1 = 0.9$, $\beta_2 = 0.999$) and a cosine decay schedule. During training, we apply random cropping to $128 \times 128$ patches, along with horizontal and vertical flipping as data augmentation. All experiments are conducted on a single NVIDIA H200 GPU (140 GB). Memory usage is approximately 42 GB for the Tiny (*i.e.*, MIRAGE -T) model and 56 GB for the Small model (*i.e.*, MIRAGE -S).

**Model Scaling.** We propose two scaled variants of our MIRAGE , namely Tiny (MIRAGE -T) and Small (MIRAGE -S). As detailed in Tab. A, these variants differ in terms of the number of MDAB blocks across scales, the input embedding dimension, the FFN expansion factor, and the number of refinement blocks.

## A.3 OPTIMIZATION OBJECTIVES

The overall optimization objective of our approach is defined as:

$$\mathcal{L}_{\text{total}} = \mathcal{L}_1 + \lambda_{fre} \times \mathcal{L}_{\text{Fourier}} + \lambda_{ctrs} \times \mathcal{L}_{\text{SPD}}. \tag{A}$$

Here, $\mathcal{L}_{\text{Fourier}}$ denotes the real-valued Fourier loss computed between the restored image and the ground-truth image, and $\mathcal{L}_{\text{SPD}}$ represents our proposed contrastive learning objective in the SPD (Symmetric Positive Definite) space.

Specifically, we adopt an $\ell_1$ loss that adopted in IR tasks Potlapalli et al. (2024); Zamfir et al. (2025); Li et al. (2022); Ren et al. (2026); Cui et al. (2025); Ren et al. (2024); Li et al. (2025), defined as $\mathcal{L}_1 = |\hat{x} - x|_1$, to enforce pixel-wise similarity between the restored image $\hat{x}$ and the ground-truth image $x$. $\mathcal{L}_{\text{Fourier}}$, as utilized in MoCE-IR Zamfir et al. (2025); Cui et al. (2025), to enhance frequency-domain consistency, the real-valued Fourier loss, is defined as:

$$\mathcal{L}_{\text{Fourier}} = \left\| \mathcal{F}_{\text{real}}(\hat{x}) - \mathcal{F}_{\text{real}}(x) \right\|_1 + \left\| \mathcal{F}_{\text{imag}}(\hat{x}) - \mathcal{F}_{\text{imag}}(x) \right\|_1, \tag{B}$$

where $\hat{x}$ and $x$ denote the restored and ground-truth images, respectively. $\mathcal{F}_{\text{real}}(\cdot)$ and $\mathcal{F}_{\text{imag}}(\cdot)$ represent the real and imaginary parts of the 2D real-input FFT (*i.e.*, rfft2). The final loss is computed as the $\ell_1$ distance between the real and imaginary components of the predicted and target frequency

spectra. Same as MoCE-IR Zamfir et al. (2025), $\lambda_{fre}$ is set to 0.1 throughout our experiments. Meanwhile, the $\mathcal{L}_{\text{SPD}}$ is defined as in Eq. 3-5 of our main manuscript. More ablation studies regarding the proposed $\mathcal{L}_{\text{SPD}}$ are provided in Sec. C.3. The temperature parameter $\tau$ of the proposed $\mathcal{L}_{\text{SPD}}$ is set to 0.1 throughout all the experiments.

## B  PRELIMINARIES ON SPD-BASED FEATURE STATISTICS

This section provides a brief background on the concepts involved in our cross-layer alignment strategy. The intention is to supply intuitive context—rather than additional derivations—for second-order feature statistics, the SPD structure, and depth-asymmetric representations used in Sec. 4.2.

**Second-order feature statistics.** Raw activations capture local appearance, but the way channels vary together often reveals more stable information about degradations. For a feature matrix $X \in \mathbb{R}^{C \times N}$, the covariance

$$\mathbf{C} = \frac{1}{N-1}(X - \mu)(X - \mu)^{\top}$$

summarizes inter-channel relationships. Diagonal entries reflect each channel's variability, while off-diagonal entries describe redundancy and dependence patterns. These structures differ consistently across layers and degradations (Fig. 4; Appendix Fig. B), making covariance a compact and informative descriptor.

**SPD property of covariance matrices.** Covariance matrices are symmetric and positive definite by construction and therefore lie in the SPD set. This structure encodes meaningful geometric information: eigenvalues represent correlation strengths, and the matrix as a whole can be interpreted as a "shape" in channel space. Preserving this structure is important—direct Euclidean operations may flatten or distort correlation patterns, an effect also reflected in the collapse observed with Euclidean contrastive learning (Fig. 6).

**Representing SPD matrices for comparison.** To compare covariance matrices within a contrastive objective, we vectorize $\mathbf{C}$ and apply a learnable projection. This retains second-order relationships while mapping them to an embedding space suitable for contrastive learning. Compared to raw feature vectors, covariance embeddings emphasize structural organization and therefore provide a more stable alignment signal.

**Depth-asymmetric representations.** Shallow and latent features naturally exhibit different statistical behavior: shallow layers respond strongly to local degradations and show pronounced redundancy, while deeper layers become more decorrelated and semantically aggregated. Their covariance matrices reflect these differences in a consistent way across degradations, making shallow–latent pairs complementary views of the same signal and a natural target for alignment.

**Intuition behind SPD-based alignment.** Aligning covariance-based SPD embeddings focuses on how channels interact, rather than on individual activation values. This yields supervision that is less sensitive to local noise and more reflective of the underlying representation structure. Encouraging shallow and latent features to share similar second-order statistics stabilizes the shared feature space required for diverse degradations.

Overall, covariance provides a compact view of channel interactions, the SPD structure preserves meaningful second-order relations, and depth-asymmetric covariance patterns naturally motivate the alignment strategy formalized in Sec. 4.2.

## C  MORE METHOD DETAILS & SUPPLEMENTARY EXPERIMENTS

### C.1  1 DEG. COMPARISON

**Single-Degradation.** In Tab. B, we compare our method against state-of-the-art approaches on single degradation tasks. For dehazing on SOTS dataset, we compare with DehazeNet Cai et al. (2016), MSCNN Ren et al. (2016), AODNet Li et al. (2017), EPDN Qu et al. (2019), FDGAN Dong et al. (2020), and all-in-one methods AirNet Li et al. (2022) and PromptIR Potlapalli et al. (2024). Our 6M parameter model achieves competitive performance (31.46 dB PSNR, 0.977 SSIM), while our 10M model establishes new state-of-the-art results (31.53 dB PSNR, 0.980 SSIM), outperforming the much

Table B: *Comparison to state-of-the-art for single degradations.* PSNR (dB, ↑) and SSIM (↑) metrics are reported on the full RGB images. **Best** performance is highlighted. Our method excels over prior works.

(a) *Dehazing*

| Method | Params. | SOTS |
|---|---|---|
| DehazeNet | - | 22.46 .851 |
| MSCNN | - | 22.06 .908 |
| AODNet | - | 20.29 .877 |
| EPDN | - | 22.57 .863 |
| FDGAN | - | 23.15 .921 |
| AirNet | 9M | 23.18 .900 |
| PromptIR | 36M | 31.31 .973 |
| MIRAGE (*Ours*) | **6M** | 31.46 .977 |
| MIRAGE (*Ours*) | 10M | **31.53 .980** |

(b) *Deraining*

| Method | Params. | Rain100L |
|---|---|---|
| DIDMDN | - | 23.79 .773 |
| UMR | - | 32.39 .921 |
| SIRR | - | 32.37 .926 |
| MSPFN | - | 33.50 .948 |
| LPNet | - | 23.15 .921 |
| AirNet | 9M | 34.90 .977 |
| PromptIR | 36M | 37.04 .979 |
| MIRAGE (*ours*) | **6M** | 37.47 .980 |
| MIRAGE (*Ours*) | 10M | **38.01 .982** |

(c) *Denoising* on BSD68

| Method | Params. | $\sigma=15$ | $\sigma=25$ | $\sigma=50$ |
|---|---|---|---|---|
| DnCNN | - | 33.89 .930 | 31.23 .883 | 27.92 .789 |
| IRCNN | - | 33.87 .929 | 31.18 .882 | 27.88 .790 |
| FFDNet | - | 33.87 .929 | 31.21 .882 | 27.96 .789 |
| BRDNet | - | 34.10 .929 | 31.43 .885 | 28.16 .794 |
| AirNet | 9M | 34.14 .936 | 31.48 .893 | 28.23 .806 |
| PromptIR | 36M | 34.34 .938 | 31.71 .897 | 28.49 .813 |
| PromptIR (Reproduce) | 36M | 34.15 .934 | 31.50 .894 | 28.33 .807 |
| MIRAGE (*ours*) | **6M** | 34.23 .936 | 31.60 .896 | 28.36 .808 |
| MIRAGE (*Ours*) | 10M | **34.25 .937** | **31.65 .898** | **28.38 .810** |

---

**Algorithm A** DynamicDepthwiseConv

---

**Require:** $\alpha \in \mathbb{R}^{B \times C \times H \times W}$        ▷ Input feature map
**Ensure:** $\alpha' \in \mathbb{R}^{B \times C \times H \times W}$        ▷ Output after dynamic depthwise conv

    **[Step 1] Generate Dynamic Kernel**
1: $K \leftarrow \texttt{AdaptiveAvgPool2D}(\alpha)$      ▷ Global context pooling
2: $K \leftarrow \texttt{Conv2D}(K, 1 \times 1, \texttt{out\_ch} = C)$      ▷ Linear projection
3: $K \leftarrow \texttt{GELU}(K)$      ▷ Non-linear activation
4: $K \leftarrow \texttt{Conv2D}(K, 1 \times 1, \texttt{out\_ch} = C \cdot k^2)$      ▷ Generate kernel weights
5: $K \leftarrow \texttt{Reshape}(K, [B \cdot C, 1, k, k])$      ▷ Form depthwise filters
    **[Step 2] Apply Depthwise Convolution**
6: $\alpha_{\text{flat}} \leftarrow \texttt{Reshape}(\alpha, [1, B \cdot C, H, W])$      ▷ Prepare for grouped conv
7: $\alpha'_{\text{flat}} \leftarrow \texttt{Conv2D}(\alpha_{\text{flat}}, K, \texttt{groups} = B \cdot C, \texttt{padding} = k \div 2)$      ▷ Apply dynamic depthwise conv
8: $\alpha' \leftarrow \texttt{Reshape}(\alpha'_{\text{flat}}, [B, C, H, W])$      ▷ Reshape back to original shape
9: **return** $\alpha'$

---

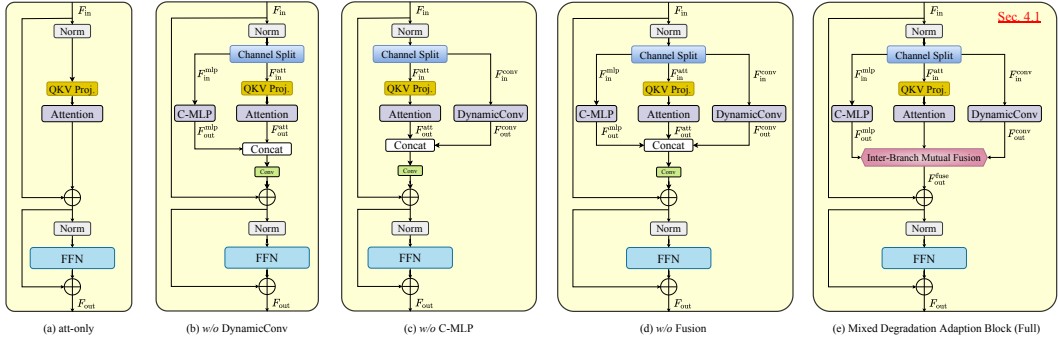

(a) att-only    (b) *w/o* DynamicConv    (c) *w/o* C-MLP    (d) *w/o* Fusion    (e) Mixed Degradation Adaption Block (Full)

Figure A: The illustration of different designs of the proposed MDAB.

larger PromptIR (36M parameters). For deraining on Rain100L, we evaluate against DIDMDN Zhang & Patel (2018), UMR Yasarla & Patel (2019), SIRR Wei et al. (2019), MSPFN Jiang et al. (2020), LPNet Gao et al. (2019), AirNet Li et al. (2022), and PromptIR Potlapalli et al. (2024). Our method significantly outperforms all baselines, with our 10M model achieving 38.01 dB PSNR and 0.982 SSIM. For denoising on BSD68, we compare with classical methods DnCNN Zhang et al. (2017a), IRCNN Zhang et al. (2017b), FFDNet Zhang et al. (2018), BRDNet Tian et al. (2020), and recent all-in-one approaches AirNet Li et al. (2022) and PromptIR Potlapalli et al. (2024). Our method consistently outperforms all competitors across different noise levels ($\sigma$=15, 25, 50), demonstrating superior performance with significantly fewer parameters than existing all-in-one methods.

**Algorithm B** SPD Contrastive Learning Optimization Pseudocode

```
# f_en: encoder
# f_de: decoder
# patch_embedding: shallow convolutional patch embedding
# refinement_conv: the refinement block and the final convolution
# spd: compute SPD feature
for x in loader: # load a minibatch x with n samples

    F_shallow = patch_embedding(x) # Convolutional Patch Embedding
    F_latent = f_en(F_shallow)

    C_s, C_l= spd(F_shallow), spd(F_latent) # Compurte SPD (Symmetric Positive Definite)
        manifold features
    z_s, z_l = proj_norm(C_s), proj_norm(C_l) # Projection and normalize

    F_recon = f_de(F_latent)
    x̂ = refinement_conv(F_recon)

    L = L_1(x, x̂) + λ_fre×L_Fourier (x, x̂) + λ_ctrs×L_SPD(z_s, z_l) # total loss

    L.backward() # back-propagate
    update(f_en, f_de, patch_embedding, refinement_conv) # SGD update

def L_Fourier(a, b): # Real-valued Fourier loss

    Please refer to Eq.B of our Appendix.

    return loss

def L_SPD(a, b): # SPD Loss

    Please refer to Eq.5 of our main manuscript.

    return loss
```

## C.2 Details of the Design for the proposed Mixed Backbone.

To investigate the effectiveness of combining MLP, convolution, and attention mechanisms, we conducted an extensive design-level ablation study. The quantitative results are presented in Tab. 7 of the main manuscript. Here, we provide detailed visual illustrations of each design in Fig. A.

**C-MLP.** To strengthen channel-wise representation, we introduce a Channel-wise MLP module, denoted as C-MLP(). Given the input feature map $F_{\text{in}}^{\text{mlp}} \in \mathbb{R}^{B \times C \times H \times W}$, we first flatten the spatial dimensions to obtain a sequence $F_{\text{in}}^{\text{mlp}} \in \mathbb{R}^{B \times C \times L}$, where $L = H \times W$. The C-MLP is implemented using two 1D convolutional layers with a GELU activation in between. The GELU function introduces non-linearity, enabling the model to learn more complex and expressive channel-wise transformations. After processing, the output is reshaped back to the original spatial format, yielding $F_{\text{out}}^{\text{mlp}} \in \mathbb{R}^{B \times C \times H \times W}$.

**Dynamic Depthwise Convolution.** The DynamicDepthwiseConv() module is designed to capture content-adaptive local structures and is employed in Alg.1 of our main manuscript. As detailed in Alg. A, the input feature $\alpha \in \mathbb{R}^{B \times C \times H \times W}$ is first passed through a global average pooling and two $1 \times 1$ convolutions to generate a dynamic depthwise kernel for each channel and sample. The input is reshaped and convolved with the generated kernels using grouped convolution, enabling sample-specific spatial filtering. The resulting output $\alpha'$ maintains the original resolution while embedding adaptive local information.

## C.3 Details of the Proposed SPD Contrastive Learning.

As shown in Alg. B, our SPD-based contrastive learning aims to align shallow and latent representations by operating in the space of symmetric positive definite (SPD) matrices. Specifically, given the shallow features extracted from the convolutional patch embedding and the latent features produced by the encoder, we compute their second-order channel-wise statistics to obtain SPD representations. These matrices are then vectorized and projected through learnable MLP layers, followed by $\ell_2$ normalization to form contrastive embeddings. An InfoNCE-style loss is applied between the shallow and latent embeddings to encourage structural alignment across depth. This contrastive term complements the pixel-level and frequency-based objectives, promoting more discriminative and consistent feature learning without introducing any additional cost during inference. Importantly,

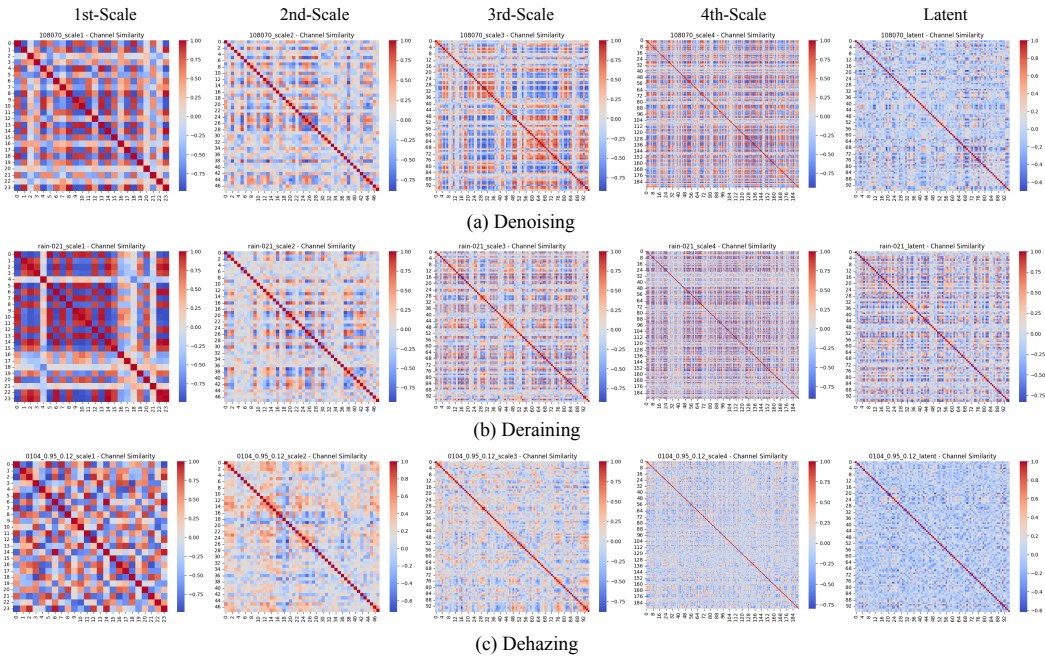

Figure B: The cross-sclae channel-wise similarity matrix visualization for Denoising, Deraining, and Dehazing.

by leveraging the geometry of second-order feature statistics, our approach implicitly regularizes the representation space, encouraging intra-instance compactness and inter-degradation separability. This geometrically grounded formulation bridges low-level signal priors with high-level contrastive learning, offering a principled and scalable solution to all-in-one image restoration.

## C.4 ABLATION REGARDING THE OPTIMIZATION OBJECTIVES

Tab. C shows that replacing SPD-based contrastive learning with a standard Euclidean-space contrastive loss (*w/o SPD*) results in a clear performance drop, demonstrating the advantage of modeling second-order channel correlations on the SPD manifold rather than relying solely on first-order vector similarities. When the entire contrastive module is removed (*w/o CL & SPD*), performance degrades even further, indicating that aligning shallow and deep features is essential for effective representation learning. Moreover, removing the Fourier loss (*w/o Fourier Loss*) slightly reduces performance, suggesting that frequency-domain supervision provides additional benefits. Overall, the full model achieves the best results, confirming the effectiveness of jointly optimizing spatial, frequency, and SPD-manifold-based structural consistency. Note that throughout all the experiments, we set $\lambda_{ctrs} = 0.05$ and $\lambda_{ctrs}=0.1$.

Table C: *Ablation Study* of MIRAGE -T on 3 Degradation Setting.

| Ablaton | Parms. | Results | |
|---|---|---|---|
| | | PSNR (dB, ↑) | SSIM(↓) |
| *w/o* CL & SPD | 5.80M | 32.63 (-0.14) | .916 |
| *w/o* SPD | 6.10M | 32.53 (-0.24) | .914 |
| *w/o* Fourier Loss | 5.80M | 32.70 (-0.07) | .917 |
| MIRAGE -T *(Full)* | 6.21M | 32.77 | .919 |

## C.5 SHALLOW-LATENT FEATURE SIMILARITY

Besides the channel-wise similarity comparison provided in our main manuscript for denoising. We also find consistent findings in other degradation, *i.e.*, raining and hazing. The corresponding channel-wise similarity across scales is provided in Fig. B. These observations reveal several important trends: *(i)* Despite the diversity of degradation types, a consistent pattern emerges across scales. Specifically, from the first to the fourth scale, the overall channel-wise similarity indicates substantial redundancy among feature channels. After channel reduction, the latent features become more decorrelated, which validates the rationale for applying contrastive learning between the latent and shallow (*i.e.*, first-scale) features. *(ii)* Different degradation types exhibit varying degrees of channel redundancy.

Table D: Zero-shot evaluation on real-world under-display camera datasets TOLED and POLED (Zhou et al., 2021).

| Method | TOLED (PSNR / SSIM / LPIPS) | POLED (PSNR / SSIM / LPIPS) |
|---|---|---|
| AirNet (Li et al., 2022) | 14.58 / 0.609 / 0.445 | 7.53 / 0.350 / 0.820 |
| PromptIR (Potlapalli et al., 2024) | 16.70 / 0.688 / 0.422 | 13.16 / 0.583 / 0.619 |
| DiffUIR (Zheng et al., 2024a) | **29.55 / 0.887 / 0.281** | 15.62 / 0.424 / 0.505 |
| **MIRAGE-S (Ours)** | 28.01 / 0.881 / 0.293 | **16.93 / 0.604 / 0.500** |

As illustrated in Fig. B, hazy images tend to produce more inherently independent features, whereas rain-degraded inputs show strong channel-wise redundancy even in the latent space. This suggests that degradations like haze may benefit from larger embedding dimensions to capture more expressive representations, while simpler degradations (*e.g.*, rain) can achieve effective restoration with smaller embedding sizes due to their inherently redundant structure.

These insights open up new directions for adaptive and degradation-aware model design in future research. Notably, this trend is not limited to the three representative samples shown; we observe similar patterns consistently across the dataset in a statistical sense. We plan to conduct a more comprehensive and quantitative investigation of this phenomenon in future work.

### C.6 More Generlization Evaluation

To further assess generalization beyond synthetic settings, we evaluate MIRAGE-S on the real-world TOLED and POLED under-display camera datasets (Zhou et al., 2021). As shown in Tab. D, MIRAGE-S achieves strong performance across both benchmarks. On POLED, which contains more severe signal attenuation and non-linear spatial artifacts, MIRAGE-S clearly surpasses prior methods across all three metrics, indicating robust transfer to challenging real-world degradations. On TOLED, MIRAGE-S remains competitive and delivers results close to diffusion-based DiffUIR despite its significantly lower complexity. These findings suggest that the proposed mixed-backbone architecture and SPD-based alignment maintain good stability under real sensor degradations and generalize reliably across distinct UDC hardware conditions.

## D Additional Visual Results.

### D.1 3 Degradation

Fig. C presents qualitative comparisons on representative cases of denoising, deraining, and dehazing, benchmarked against recent state-of-the-art methods. The proposed MIRAGE consistently yields more visually faithful restorations, characterized by enhanced structural integrity, finer texture details, and reduced artifacts. These results underscore the effectiveness of our unified framework in handling diverse degradation types while preserving high-frequency information and geometric consistency.

### D.2 5 Degradation

For the 5-degradation setting, we provide visual comparisons for the low-light enhancement task in Fig. D. As illustrated, the proposed MIRAGE produces noticeably cleaner outputs with improved luminance restoration and better color consistency compared to MoCE-IRZamfir et al. (2025), demonstrating its robustness under challenging illumination conditions.

### D.3 Composited Degradation

Fig. E and Fig. F present visual comparisons under more challenging composite degradations, namely *low-light + haze + snow* and *low-light + haze + rain*, respectively. As observed, our method reconstructs significantly more scene details and preserves structural consistency, whereas MoCE-IR Zamfir et al. (2025) tends to produce noticeable artifacts and over-smoothed regions under these complex conditions.

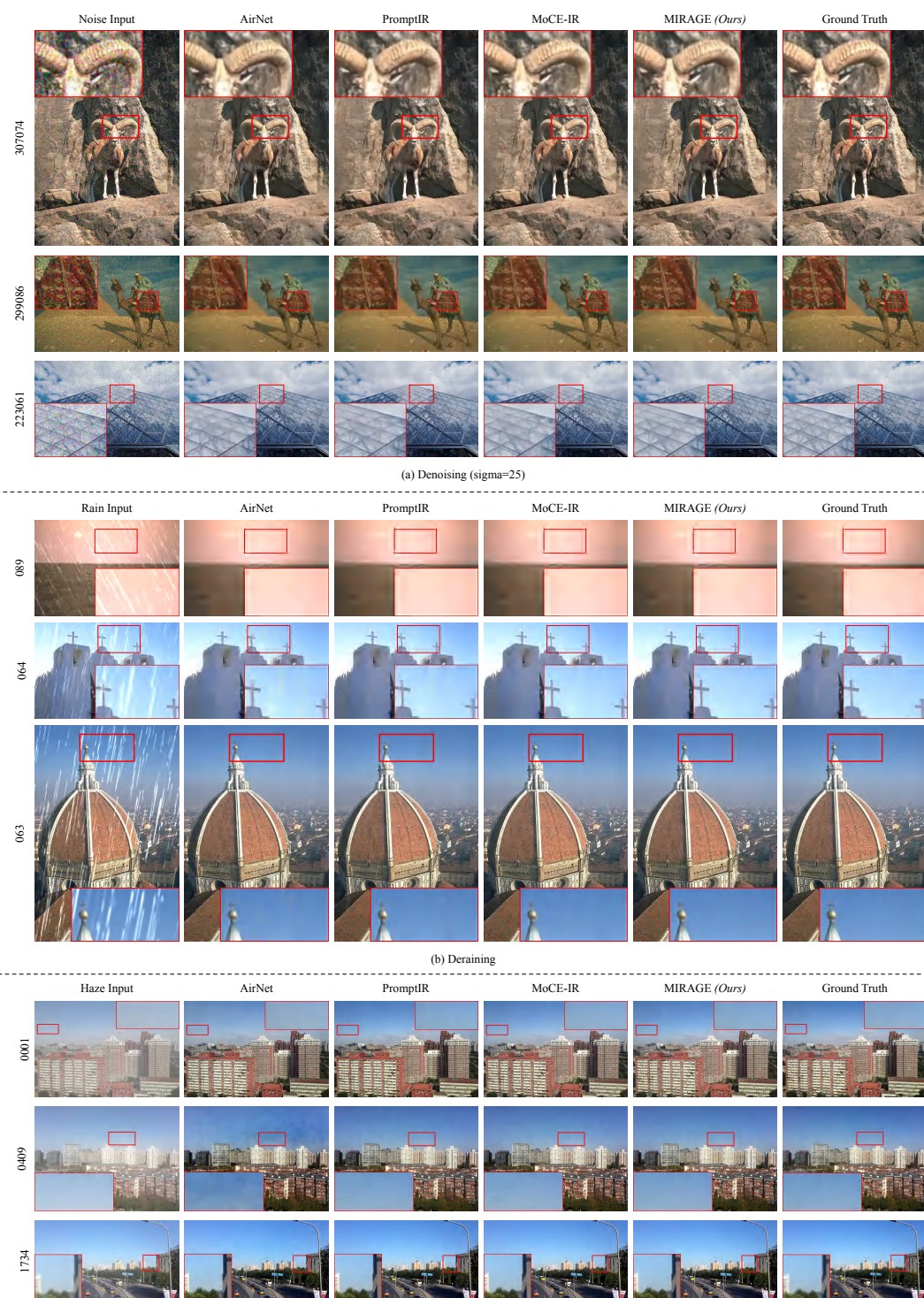

Figure C: Visual comparison of MIRAGE with state-of-the-art methods considering three degradations. Zoom in for a better view.

## D.4    ZERO-SHOT UNDERWATER IMAGE ENHANCEMENT

Fig. G demonstrates that even when directly applied to unseen underwater images, our method is able to effectively enhance visibility and contrast, producing results that are noticeably clearer than the

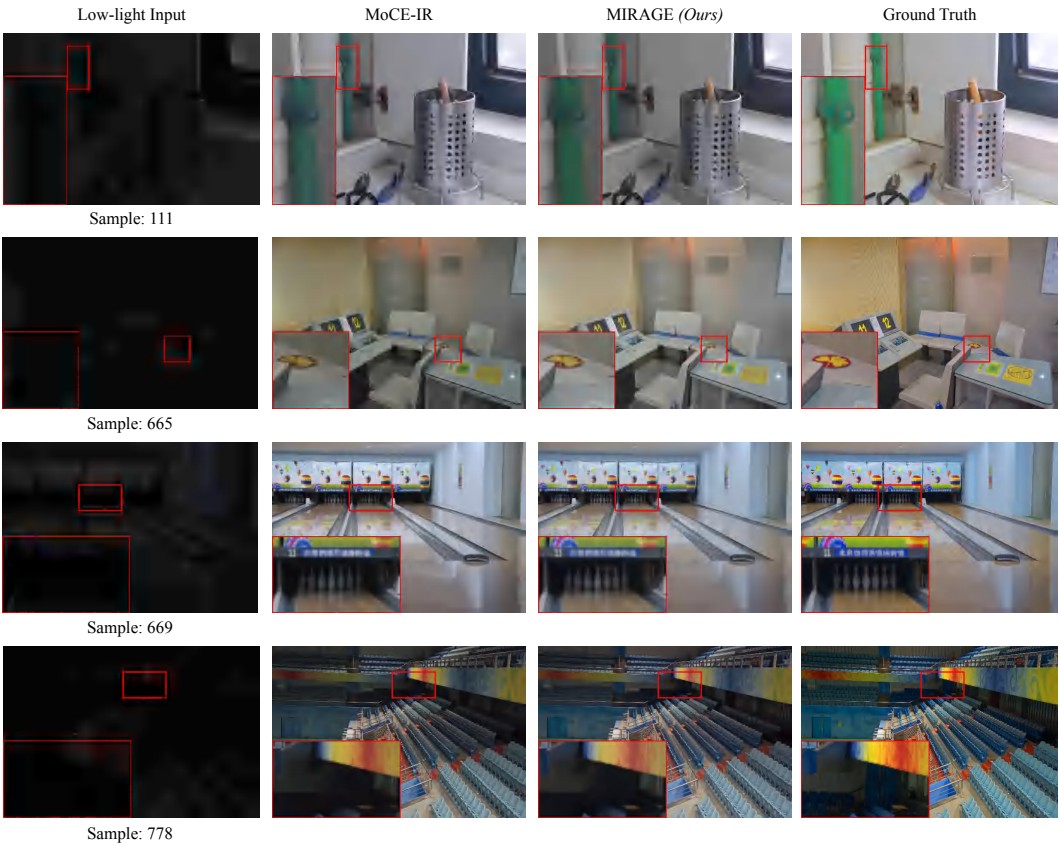

Figure D: Visual comparison of MIRAGE with state-of-the-art methods considering low-light degradation. Zoom in for a better view.

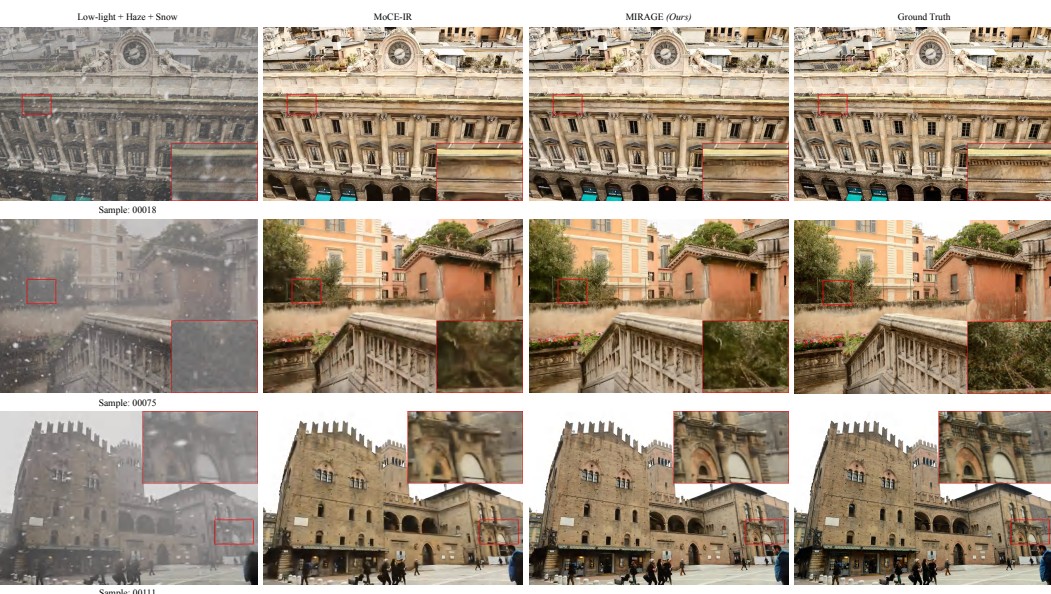

Figure E: Visual comparison of MIRAGE with state-of-the-art methods considering composited degradation (Low-light + Haze + Snow). Zoom in for a better view.

raw input and visually closer to the reference images. This qualitative evidence further validates the strong generalization ability of the proposed framework to unseen domains.

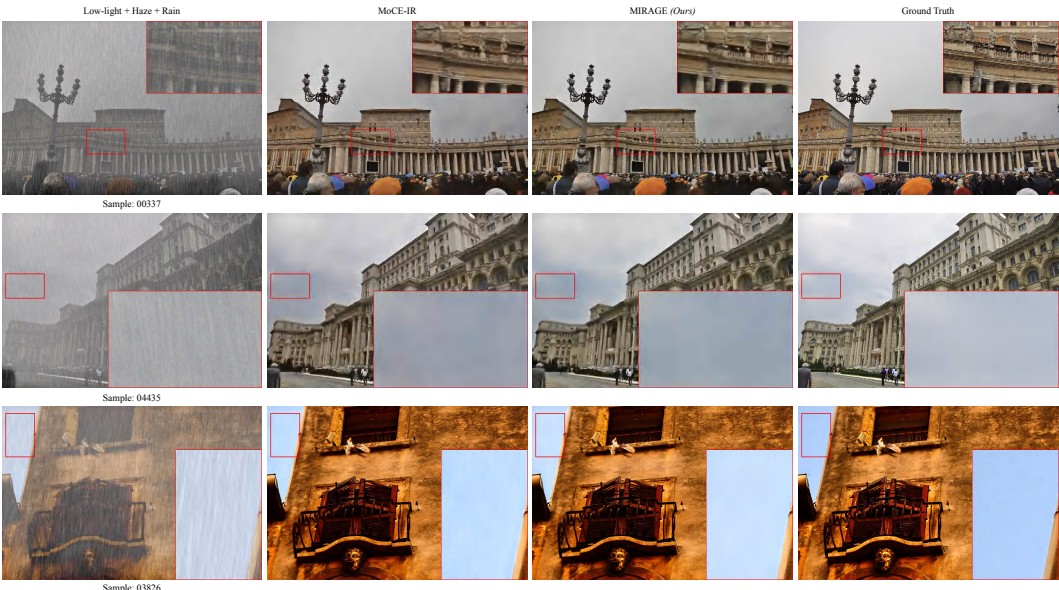

Figure F: Visual comparison of MIRAGE with state-of-the-art methods considering composited degradation (Low-light + Haze + Rain). Zoom in for a better view.

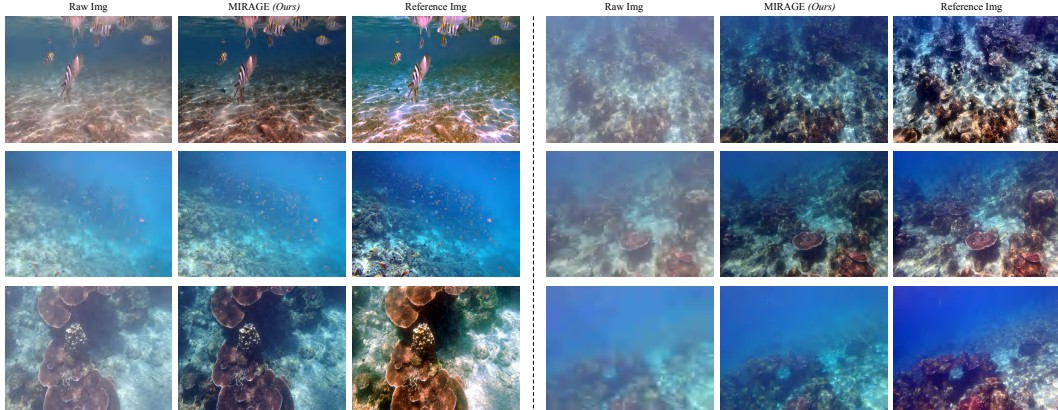

Figure G: Visual results of MIRAGE for Underwater Image Enhancement. Zoom in for a better view.

# E  LIMITATIONS AND FUTURE WORK

While the proposed MIRAGE achieves new state-of-the-art performance on most all-in-one image restoration benchmarks, we observe that its deblurring performance still lags slightly behind MoCE-IR Zamfir et al. (2025). We attribute this to the relatively compact model size of our current design, which favors efficiency over aggressive capacity. To address this, future work will explore scaling up the model size to be on par with larger architectures such as PromptIR Potlapalli et al. (2024), MoCE-IR Zamfir et al. (2025), and AdaIR Cui et al. (2025), aiming to further boost performance while maintaining the architectural elegance and efficiency of our design. Moreover, our current SPD-based contrastive learning leverages a conventional InfoNCE loss in Euclidean space after projecting SPD features. While effective, it does not fully exploit the intrinsic geometry of the SPD manifold. As part of future efforts, we plan to investigate geodesic-based contrastive formulations and Riemannian-aware optimization strategies, which may offer a more principled and theoretically grounded way to align structured representations across semantic scales. Additionally, different degradations may favor different proportions of convolution, attention, and MLP capacity. Learning such ratios dynamically is an interesting direction and could further adapt MIRAGE to degradation-specific characteristics. We view this as a promising avenue for future research.

## F    BROADER IMPACT

Image restoration (IR) is a fundamental task with applications in photography, remote sensing, surveillance, autonomous driving, medical imaging, and scientific visualization. By proposing a unified and efficient framework capable of handling diverse degradation types with minimal computational cost, our work may benefit scenarios where image quality is compromised by environmental or hardware constraints. The lightweight design of MIRAGE further enables deployment on resource-limited devices such as mobile phones, drones, or embedded cameras, which can support use cases in low-resource settings or critical domains like emergency response and environmental monitoring. From a research perspective, our modular design and SPD-based contrastive formulation may encourage further exploration of geometrically-aware representation learning in restoration and related areas.

## G    USE OF LARGE LANGUAGE MODELS (LLMS)

We used OpenAI's GPT-based Large Language Models (LLMs) (OpenAI, 2023; 2022) to polish the writing and improve the readability of the paper. The models were not used for developing the methodology, running experiments, or analyzing results. All scientific contributions remain entirely the work of the authors.

