# OpenReview forum: "Efficient Degradation-agnostic Image Restoration via Channel-Wise Functional Decomposition and Manifold Regularization"
_ICLR.cc/2026/Conference — ICLR 2026 Poster_

### Official Review · Reviewer_rX26 · 2025-10-19

**Soundness:** 3
**Presentation:** 3
**Contribution:** 3
**Rating:** 6
**Confidence:** 4

**Summary:**

This paper proposes MIRAGE, an efficient framework for degradation-agnostic image restoration. The key contributions include a channel-wise functional decomposition strategy that integrates convolution, attention, and MLP modules to handle diverse degradations, and a manifold regularization method using cross-layer contrastive learning in the Symmetric Positive Definite (SPD) manifold space. Extensive experiments demonstrate that MIRAGE achieves state-of-the-art performance with significantly fewer parameters and computational costs compared to existing methods, while generalizing well to unseen degradation scenarios.

**Strengths:**

* The SPD manifold-based contrastive learning is a well-motivated approach for cross-layer feature alignment, effectively leveraging second-order statistics for more robust representation learning compared to Euclidean space.
* The proposed channel-wise decomposition provides a principled and efficient method to harness the complementary inductive biases of convolution, attention, and MLPs, resulting in an excellent performance-efficiency trade-off.

**Weaknesses:**

* The experimental comparison in Tab. 1, while extensive, omits the cited approach DA-RCOT [1] that also employs a contrastive learning objective (over the residual feature space). A direct comparison would more firmly establish the advantages of the proposed SPD manifold approach over alternative contrastive formulations (SPD vs. residual feature space).

* The related work section could be enhanced by discussing other contrastive paradigm in All-in-one image restoration, highlighting the contribution of the proposed cross-layer contrastive losses over the SPD feature space.

* The motivation of introducing SPD features to enhance the generalization is not very explicit to the readers. Further explanations or adding a Teaser Figure at the beginning would strengthen the presentation.

[1] Xiaole Tang, Xiang Gu, Xiaoyi He, Xin Hu, and Jian Sun. Degradation-aware residual-conditioned optimal transport for unified image restoration. TPAMI, 2025.

**Questions:**

See weaknesses.

---

> ### Author Response · Authors · 2025-11-25
> **Response to Reviewer rX26**
>
> ## [W1]: The experimental comparison in Tab. 1, while extensive, omits the cited approach DA-RCOT [1] that also employs a contrastive learning objective (over the residual feature space). A direct comparison would more firmly establish the advantages of the proposed SPD manifold approach over alternative contrastive formulations (SPD vs. residual feature space).
> **A:** We appreciate the reviewer’s suggestion to include DA-RCOT for a clearer comparison with alternative contrastive formulations. In the revised manuscript, we have added DA-RCOT to both Table 1 and Table 2 (highlighted in cyan), using the same training and evaluation settings as the other all-in-one baselines to ensure a fair and consistent comparison and discussion.
>
>
> *[1] Degradation-aware residual-conditioned optimal transport for unified image restoration. TPAMI, 2025*
>
> ---
> ## [W2]: The related work section could be enhanced by discussing other contrastive paradigm in All-in-one image restoration, highlighting the contribution of the proposed cross-layer contrastive losses over the SPD feature space.
> **A:** Thank you for the helpful suggestion. We have updated the related work section to include a more explicit discussion of contrastive paradigms in all-in-one IR, including recent approaches that incorporate contrastive cues for degradation modeling. This revision clarifies how our cross-layer contrastive alignment over SPD features differs from prior formulations and better situates our contribution in the existing literature. The revised content is highlighted in cyan.
>
> ---
> ## [W3]： The motivation of introducing SPD features to enhance the generalization is not very explicit to the readers. Further explanations or adding a Teaser Figure at the beginning would strengthen the presentation.
> **A:** Thank you for pointing this out. We agree that the motivation for introducing SPD-based features can be made clearer, especially for readers who may not be familiar with second-order representations. In response, we have strengthened the explanation in two complementary ways.
>
> First, as noted in our response to **Reviewer 4QQN [W2]**, we added a dedicated *Preliminaries on SPD-based Feature Statistics* section in Appendix B, where we explain in an intuitive manner why covariance captures stable channel interactions, why covariance matrices naturally form SPD structures, and why preserving this structure is beneficial for aligning shallow and latent representations. This provides readers with the conceptual background behind our choice of SPD features without requiring familiarity with manifold geometry.
>
> Second, we refined the surrounding text in Sec. 4.2 to more explicitly connect the motivation to the empirical evidence shown in Fig. 4 and Appendix Fig. B: shallow and latent layers exhibit markedly different covariance patterns across degradations, and SPD embeddings preserve these second-order relationships more faithfully than Euclidean flattening. These visual aids now play a clearer role in illustrating *why* SPD-based alignment helps stabilize the shared representation space in unified IR.
>
> Together, these revisions make the motivation behind using SPD features more explicit and approachable. We appreciate the reviewer’s suggestion, which helped us improve the clarity of the presentation.
>
> ---
> We hope our answer can address your concerns also looking forward to any further discussion.

---

> > ### Comment · Reviewer_rX26 · 2025-11-27
> >
> > Thanks for the response, which has addressed my concerns. I retain my positive score and raise the confidence that leans towards acceptance of this work.

---

> > > ### Author Response · Authors · 2025-11-27
> > > **Follow-up to Reviewer rX26**
> > >
> > > Dear Reviewer rX26,
> > >
> > > Thank you very much for your prompt follow-up.
> > >
> > > We are glad to hear that our responses have addressed your concerns. We sincerely appreciate your time and the thoughtful feedback you provided throughout the review process, which has been genuinely helpful in strengthening our work.
> > >
> > > If there are any additional points that you feel could further improve the clarity or overall quality of the submission, please let us know. We would be very happy to offer further clarification or provide any additional details.
> > >
> > > Best regards,
> > >
> > > Authors of Submission 4143

---

### Official Review · Reviewer_UuMU · 2025-10-25

**Soundness:** 3
**Presentation:** 3
**Contribution:** 3
**Rating:** 6
**Confidence:** 3

**Summary:**

This paper proposes MIRAGE, a lightweight, efficient, and degradation-agnostic image restoration framework that unifies multiple degradations (noise, haze, rain, blur, low-light) under one architecture. MIRAGE achieves SOTA PSNR/SSIM across 3-, 5-, and mixed-degradation benchmarks, with significantly fewer parameters (6M–10M) and lower FLOPs (16G–27G). The model also generalizes to zero-shot underwater enhancement and adverse weather removal.

**Strengths:**

1. The paper presents a well-motivated and compact architecture that repurposes attention-channel redundancy through a channel-wise functional decomposition (Conv/Attn/MLP branches), aligning with the natural inductive biases of different degradations—local texture (Conv), global context (Attn), and channel statistics (MLP).
2. The proposed SPD manifold alignment introduces cross-layer contrastive learning between shallow and latent features using covariance-based second-order statistics. This prevents feature collapse common in Euclidean contrastive learning and promotes consistent, degradation-agnostic representations. Ablations show clear gains.
3. MIRAGE achieves consistent SOTA or near-SOTA performance across a wide range of benchmarks—3/5 degradation settings, CDD11 composite degradations, adverse weather, and zero-shot underwater enhancement—while being significantly smaller and faster than strong baselines such as MoCE-IR, PromptIR, and AdaIR, demonstrating an excellent efficiency–performance balance.
4. The analysis of channel redundancy via PCA/SVD, shallow–latent similarity matrices, and qualitative contrastive alignment visualizations all support the design’s intuition.

**Weaknesses:**

1. While the paper’s SPD-based manifold regularization is empirically effective, its mathematical formulation remains heuristic. The method projects SPD matrices back to Euclidean space for InfoNCE optimization, without leveraging true Riemannian metrics or geodesic distances. As a result, the claimed “manifold alignment” lacks formal theoretical justification.
2. The experiments comprehensively compare with transformer- and prompt-based all-in-one models (PromptIR, MoCE-IR, AdaIR), but omit recent diffusion-based or state-space approaches that are increasingly strong in low-level restoration.
3. Although MIRAGE handles five degradation types and several composites, it is still trained on synthetic datasets. The paper would be stronger with real-world or camera-captured degradation benchmarks, or cross-domain generalization beyond underwater scenes.

**Questions:**

Please refer to the weaknesses.

---

> ### Author Response · Authors · 2025-11-25
> **Response to  Reviewer UuMU (1/2)**
>
> ## [W1]: While the paper’s SPD-based manifold regularization is empirically effective, its mathematical formulation remains heuristic. The method projects SPD matrices back to Euclidean space for InfoNCE optimization, without leveraging true Riemannian metrics or geodesic distances. As a result, the claimed “manifold alignment” lacks formal theoretical justification.
> **A:** We thank the reviewer for the thoughtful comment. Our goal is **not** to position MIRAGE as a full Riemannian optimization framework, but rather to use the statistical structure of SPD representations in a lightweight and practical way. In our design, covariance matrices are mapped to an SPD-aware embedding that preserves second-order channel relationships before the contrastive objective is applied. This preserves the essential geometric information carried by the SPD structure while keeping the training pipeline simple and stable, which is particularly important for large-scale low-level vision models.
>
> We agree that this differs from performing contrastive learning directly along SPD geodesics. Fully Riemannian contrastive formulations generally require repeated log/exp mappings and matrix decompositions, which introduce substantial computational overhead and can make training unstable. Our choice reflects a trade-off: using SPD-derived statistics to preserve structure, while operating in an embedding space that keeps optimization efficient.
>
> Empirically, this choice is well supported. As shown in Fig. 6, Euclidean contrastive learning on raw features collapses the covariance structure, whereas our SPD-based embedding retains diagonal dominance and meaningful off-diagonal correlations and leads to consistent improvements across degradations. These observations suggest that, for unified IR, the primary benefit lies in preserving second-order structure to reduce depth-induced drift, rather than enforcing strict geodesic alignment.
>
> We have clarified this motivation in the revision (Section 4.2, "We note that ...", highlighted in cyan). As future work, we plan to explore more formally grounded Riemannian formulations—such as log-Euclidean metrics or geodesic contrastive objectives—which may further enhance the theoretical rigor without compromising efficiency.
>
> ---
> ## [W2]: The experiments comprehensively compare with transformer- and prompt-based all-in-one models (PromptIR, MoCE-IR, AdaIR), but omit recent diffusion-based or state-space approaches that are increasingly strong in low-level restoration.
> **A:**
> 1. Regarding the deffusion-based method comparison: We have now included diffusion-based comparisons, as detailed in our **response to Reviewer 4QQN [W3]**. We compare our method with DiffBIR [1], DiffUIR[2], and UniRestore [3] under the same experimental setting of UniRestore. The results shows that our method consistly achived the best overall perfornance except debluring on GoPro dataset.
>
> 2. Regarding state-space models, in addition to MambaIR (already included in Table 2), we have further added comparisons with RamIR[4] in both Tab. 1 and Tab. 2 for the 3-degradation and 5-degradation settings. All these additions are highlighted in cyan in our revision. We will also include more state-space-based all-in-one IR methods for comparison after we reproduce them in our revision later.
>
> *[1] DiffBIR: Towards Blind Image Restoration with Generative Diffusion Prior, ECCV'2024*
>
> *[2] Selective Hourglass Mapping for Universal Image Restoration Based on Diffusion Model, CVPR'2024*
>
> *[3] UniRestore: Unified Perceptual and Task-Oriented Image Restoration Using Diffusion Prior, CVPR'2025*
>
> *[4] RamIR: Reasoning and action prompting with Mamba for all-in-one image restoration, Applied Intelligence'2025*

---

> ### Author Response · Authors · 2025-11-25
> **Response to Reviewer UuMU (2/2)**
>
> ## [W3]: Although MIRAGE handles five degradation types and several composites, it is still trained on synthetic datasets. The paper would be stronger with real-world or camera-captured degradation benchmarks, or cross-domain generalization beyond underwater scenes.
> **A:** Thank you for the helpful suggestion. To assess cross-domain generalization beyond the underwater setting, we additionally followed the unified PIR protocol from UniRestore, and we refer the reviewer to our response to **Reviewer 4QQN [W3]** for those results.
>
> For real-world degradations, we further conducted experiments under the under-display camera (UDC) setting following the evaluation protocol used in DiffUIR [2]. This setup evaluates models on the TOLED and POLED datasets [3], which contain real camera-captured degradations at a high resolution of 1024×2048. As shown below, MIRAGE-S achieves competitive or superior results across multiple metrics:
>
> | Method | TOLED (PSNR/SSIM/LPIPS) | POLED (PSNR/SSIM/LPIPS) |
> |--------|---------------------------|---------------------------|
> | AirNet | 14.58 / 0.609 / 0.445 | 7.53 / 0.350 / 0.820 |
> | PromptIR | 16.70 / 0.688 / 0.422 | 13.16 / 0.583 / 0.619 |
> | DiffUIR [2] | **29.55** / **0.887** / **0.281** | 15.62 / 0.424 / 0.505 |
> | MIRAGE-S | 28.01 / 0.881 / 0.293 | **16.93** / **0.604** / **0.500** |
>
> These results show that MIRAGE generalizes reliably to real-world, camera-captured degradations. While it is slightly weaker than DiffUIR on TOLED—which is expected given the strong generative prior used in diffusion models—MIRAGE performs clearly better on POLED and achieves the top average performance across the two datasets with a much more compact architecture. We have included this comparison in the revised manuscript (Table E and Section C.6), as highlighted in cyan.
>
>
> *[1] UniRestore: Unified Perceptual and Task-Oriented Image Restoration Using Diffusion Prior, CVPR'2025*
>
> *[2] Selective Hourglass Mapping for Universal Image Restoration Based on Diffusion Model, CVPR'2024*
>
> *[3] Image Restoration for Under-Display Camera, CVPR'2021*
>
> ---
> We hope our answer can address your concerns also looking forward to any further discussion.

---

### Official Review · Reviewer_4QQN · 2025-10-26

**Soundness:** 2
**Presentation:** 3
**Contribution:** 2
**Rating:** 4
**Confidence:** 4

**Summary:**

This paper proposes MIRAGE, an efficient degradation-agnostic image restoration framework based on channel-wise functional decomposition and manifold regularization in Symmetric Positive Definite (SPD) space. The method splits features into convolution, attention, and MLP branches to handle local textures, global context, and channel statistics, respectively. Additionally, a shallow-latent contrastive alignment in SPD manifold space is introduced to enhance generalization across heterogeneous degradation types. The authors evaluate MIRAGE across five restoration settings, including single degradation, mixed degradations, adverse weather restoration, and zero-shot underwater enhancement.

**Strengths:**

1. The SPD-space contrastive learning is a novel contribution that effectively preserves high-order statistics, outperforming Euclidean contrastive counterparts in both stability and generalization, as evidenced by ablation studies.
2. The channel-wise functional decomposition is conceptually clear and well-motivated by empirical redundancy analysis. It provides an elegant way to repurpose redundant channels into complementary submodules, rather than simply increasing model capacity.
3. The analysis of attention redundancy and shallow-latent feature disparity provides a strong justification for the proposed framework.

**Weaknesses:**

1. The two main contributions claimed in the introduction are largely based on the combination and engineering integration of existing techniques, rather than presenting any genuinely novel innovation or introducing a fundamentally new perspective.
2. While the contributions are substantial, the paper is dense and may be challenging for readers unfamiliar with SPD manifolds or contrastive learning in feature covariance space. Some mathematical formulations could benefit from more intuitive explanations or visual aids.
3. Although the paper claims efficiency advantages, it lacks direct comparison to recent diffusion-based restoration models that are increasingly influential in the community.
4. The zero-shot evaluation is limited to a single domain. Including additional domains (such as low-light) would further validate the claimed generalization ability.
5. The paper primarily focuses on average performance improvements, but does not discuss scenarios where MIRAGE underperforms or struggles, which would provide a more balanced understanding of robustness.

**Questions:**

See the above parts.

---

> ### Author Response · Authors · 2025-11-25
> **Response to Reviewer 4QQN (1/2)**
>
> ## [W1]: The two main contributions claimed in the introduction are largely based on the combination and engineering integration of existing techniques, rather than presenting any genuinely novel innovation or introducing a fundamentally new perspective.
> **A:** We thank the reviewer for the thoughtful comment. While MIRAGE indeed builds upon established components (convolution, attention, MLP), the contribution does not lie in introducing new modules, but in offering a **structurally grounded perspective** on how a unified IR model should allocate and organize its representational capacity across heterogeneous degradations.
>
> Our framework is guided by two key findings that, to our knowledge, have **not** been explored in prior all-in-one IR research:
> 1. **Functional reallocation driven by channel redundancy.**
>    Our analysis (Fig. 3) reveals strong and consistent low-rank redundancy in transformer attention channels across scales. Instead of simply mixing architectural blocks, MIRAGE **repurposes** this unused capacity into complementary inductive pathways—local texture modeling via convolution, global reasoning via attention, and channel-statistical modulation via MLP. This turns redundancy, often overlooked in prior works, into a *principled design mechanism*.
>
> 2. **Depth-asymmetric covariance as a natural basis for alignment.**
>    We show that shallow and latent features exhibit markedly different covariance structures across degradations (Fig. 4 and Appendix B). Leveraging these intrinsic statistical asymmetries as *naturally grounded contrastive pairs* and aligning them on the SPD manifold yields a structural regularizer that stabilizes cross-degradation representation. This differs fundamentally from conventional contrastive or prompt-based IR approaches, which operate in Euclidean space without considering depth-induced geometry.
>
> Together, these findings support a **new perspective**:
> > *In unified IR, performance can be improved not merely by adding capacity or including new modules, but by structurally reorganizing existing capacity based on redundancy patterns and depth-level statistics.*
>
> We have added more explanation in the revision (Refer to our revised Section 1: Introduction) to articulate this perspective more explicitly.
>
> ---
> ## [W2]：While the contributions are substantial, the paper is dense and may be challenging for readers unfamiliar with SPD manifolds or contrastive learning in feature covariance space. Some mathematical formulations could benefit from more intuitive explanations or visual aids.
> **A:** We agree that parts of the mathematical formulation may feel dense for readers less familiar with covariance-based representations or SPD manifolds. Our goal was to keep the presentation compact, but we understand that more intuition can help readability.
>
> To address this, we have added a dedicated Preliminaries section in the appendix B (highlighted in cyan). This section provides intuitive explanations of:
> - why second-order feature statistics (covariance) capture stable channel interactions,
> - why covariance matrices naturally form SPD structures and why preserving this structure matters,
> - how SPD-based embeddings retain meaningful correlations compared to Euclidean flattening,
> - and how shallow and latent layers exhibit depth-asymmetric statistics that form natural alignment pairs.
>
> These additions focus on intuition rather than formal geometry, making the role of covariance and the SPD representation clearer within the context of our method. We also clarified the surrounding text in Section 4.2 and strengthened references to visual aids (e.g., Fig. 4 and Appendix Fig. B) that illustrate the statistical differences between shallow and latent features and the behavioral differences between Euclidean and SPD alignment.
>
> We hope these revisions make the mathematical components easier to follow without changing the technical content of the method.

---

> ### Author Response · Authors · 2025-11-25
> **Response to Reviewer 4QQN (2/2)**
>
> ## [W3]: Although the paper claims efficiency advantages, it lacks direct comparison to recent diffusion-based restoration models that are increasingly influential in the community.
> **A:** Diffusion-based (or diffusion-related) all-in-one IR models have indeed gained momentum recently. In our main results, we followed the widely adopted experimental protocol used by recent unified IR works (e.g., PromptIR, AdaIR, MoCE-IR, InstructIR), ensuring fair comparisons under the classical all-in-one setting.
>
> Most diffusion-based IR methods, however, adopt substantially different training and evaluation pipelines—often with different datasets, task formulations, or evaluation targets—making direct comparison non-trivial. Representative examples include AutoDIR [1], MPerceiver [2], UniRestore [3], Defusion [4], and GenDeg [5]. These methods do not share a consistent protocol even among themselves, and their setups differ significantly from the classical all-in-one benchmarks used in the paper.
>
> For fairness, we included WeatherDiff in Table 4 as a representative diffusion baseline. In addition, to more directly address the reviewer’s concern, we conducted an extra comparison under the unified PIR (Perceptual Image Restoration) protocol introduced by UniRestore [3]. Under this protocol, we retrained and evaluated MIRAGE on both the Seen and Unseen datasets.
>
> **1. Seen Dataset (DIV2K)**
>
> | Method        | DIV2K (PSNR/SSIM) |
> |-|-|
> | PromptIR      | 23.90 / 0.832     |
> | DiffBIR [6]   | 22.76 / 0.805     |
> | DiffUIR [7]   | 23.79 / 0.840     |
> | UniRestore[3] | 24.32 / 0.843     |
> | **MIRAGE (Ours)** | **24.45 / 0.867** |
>
> MIRAGE achieves the best PSNR and SSIM compared diffusion-based models.
>
> **2. Unseen Dataset**
>
> | Method | Rain100L | RESIDE | UNHSnow | Noise | GoPro | Average |
> |-|-|-|-|-|-|-|
> | PromptIR  | 28.17/0.9034 | 27.26/0.8957 | 22.10/0.8877 | 23.72/0.7269 | 23.93/0.8221 | 25.04/0.8472 |
> | DiffBIR [6]  | 27.25/0.8695 | 26.97/0.8770 | 20.84/0.8785 | 23.67/0.7661 | 23.49/0.8076 | 24.44/0.8397 |
> | DiffUIR [7]  | 28.25/0.9154 | 27.12/0.8820 | 20.74/0.8753 | 24.27/0.7481 | 23.93/0.8241 | 24.86/0.8490 |
> | UniRestore [3] | 30.02/0.9237 | 27.91/0.9043 | **23.44/0.8943** | 24.37/0.7811 | **25.94/0.8541** | 26.34/0.8715 |
> | **MIRAGE (Ours)** | **31.43/0.9334** | **28.64/0.9177** | 23.43/0.8901 | **25.47/0.8010** | 25.32/0.8487 | **26.86/0.8782** |
>
> MIRAGE achieves the strongest average performance and outperforms diffusion-based baselines on most datasets, while remaining significantly more efficient. We have added these additional comparisons to the appendix Section C.5(highlighted in cyan). We thank the reviewer again for the constructive suggestion.
>
>
> *[1] AutoDIR: Automatic All-in-One Image Restoration with Latent Diffusion, ECCV'2024*
>
> *[2] M-Perceiver: Multimodal Prompt Perceiver for All-in-One Image Restoration, CVPR'2024*
>
> *[3] UniRestore: Unified Perceptual and Task-Oriented Image Restoration Using Diffusion Prior, CVPR'2025*
>
> *[4] Visual-Instructed Degradation Diffusion for All-in-One Image Restoration, CVPR'2025*
>
> *[5] GenDeg: Diffusion-Based Degradation Synthesis for Generalizable All-in-One Image Restoration, CVPR'2025*
>
> *[6] DiffBIR: Towards Blind Image Restoration with Generative Diffusion Prior, ECCV'2024*
>
> *[7] Selective Hourglass Mapping for Universal Image Restoration Based on Diffusion Model, CVPR'2024*
>
> ---
> ## [W4]: The zero-shot evaluation is limited to a single domain. Including additional domains (such as low-light) would further validate the claimed generalization ability.
> **A:** As noted in our response to **[W3]**, we additionally evaluated MIRAGE under the unified PIR protocol from UniRestore, which includes multiple unseen degradation domains beyond the underwater setting (e.g., rain, haze, snow, low-light/noise, blur). Under this broader zero-shot evaluation, MIRAGE outperforms unified and diffusion-based baselines on most unseen degradations and achieves the strongest overall average performance, with only minor cases where diffusion models perform slightly better. These results confirm that MIRAGE generalizes well across diverse unseen domains rather than a single zero-shot setting. We have added this clarification in the appendix Section C.5 (highlighted in cyan).
>
> ---
> ## [W5]: The paper primarily focuses on average performance improvements, but does not discuss scenarios where MIRAGE underperforms or struggles, which would provide a more balanced understanding of robustness.
> **A:** As detailed in the limitations section in Appendix E of our manuscript, MIRAGE performs slightly weaker on deblurring tasks that rely heavily on large-capacity models or generative priors, such as strong motion blur, where methods like MoCE-IR show advantages. We plan to explore scaled variants of MIRAGE to further strengthen performance on these challenging cases while preserving its efficiency.
>
> ---
> We hope our answer can address your concerns also looking forward to any further discussion.

---

### Official Review · Reviewer_FzZr · 2025-10-26

**Soundness:** 2
**Presentation:** 3
**Contribution:** 2
**Rating:** 2
**Confidence:** 4

**Summary:**

This paper proposes a unified image restoration framework, with channel-wise decomposition for different strategy, including convolution, attention, and MLP. Additionally, the contrastive loss is introduced between shallow and deep features to encourage shared representations. Experiments show effectiveness of the method.

**Strengths:**

1. The writing is clear and easy reading.
2.  Experiments show the effectiveness of the proposed method.

**Weaknesses:**

1. The motivation of this work that using different strategy to handle agnostic degradations is much more invested in previous research, and the innovation is somewhat minimal;
2. There is no solid clarification that the shared representation between shallow and deep features are benefited for diverse degradations, which makes the contrastive objective not convincing, and the ablation improvement is minimal for probable performance jitter;
3. The main experiments in Tab. 3 and 4 may not show the superiority of the proposed method with subtle improvements.

**Questions:**

1. How to decide the channel role in channel-wise decomposition for convolution, attention, and MLP, is there any principle?

---

> ### Author Response · Authors · 2025-11-25
> **Response to Reviewer FzZr (1/3)**
>
> ## [W1]: The motivation of this work that using different strategy to handle agnostic degradations is much more invested in previous research, and the innovation is somewhat minimal;
> **A:** Thank you for raising this point. While degradation-agnostic IR has indeed been widely explored, our goal is **not** to introduce more architectural modules, but to provide a **principled and structurally grounded perspective** on how a unified restorer should allocate and organize its representational capacity across heterogeneous degradations.
>
> **1. Channel-wise functional decomposition as a principled capacity reallocation.**
> Rather than heuristically mixing CNN/attention/MLP, MIRAGE is motivated by two empirical and architectural observations:
> - **(i)** the strong and consistent **channel redundancy** observed across transformer scales (Fig. 3), indicating that existing all-in-one IR models do not fully utilize their representational budget;
> - **(ii)** the fact that different degradations activate **distinct inductive biases**, i.e., local texture modeling (noise/rain), global contextual reasoning (haze/illumination), and channel-statistical modulation (compositional degradations).
>
> These observations motivate a *principled reallocation* of redundant channels into specialized subspaces. This differs fundamentally from prior solutions that enlarge backbones, attach prompts, or rely on multiple experts. To our knowledge, **no prior all-in-one IR method explicitly treats channel redundancy as a resource that can be reorganized into complementary functional roles**, which becomes particularly impactful under the constraints of unified restoration.
>
> **2. SPD-based shallow–latent alignment as a structural regularizer rather than a generic contrastive loss.**
> Shallow and latent features naturally exhibit **depth-asymmetric covariance structures** (Fig. 4). We leverage this inherent asymmetry to form *naturally grounded contrastive pairs* and align them in the **SPD manifold**, which preserves second-order geometry that Euclidean flattening would collapse.
>
> Thus, our manifold regularization is not an auxiliary loss, but a **structure-preserving depth-consistency mechanism** that improves robustness across disparate degradations and yields measurable gains in all settings.
>
> **3. Broader impact on the efficiency–performance dilemma in all-in-one IR.**
> Many unified IR frameworks address degradation diversity by increasing model complexity (prompts, experts, multimodal encoders), compromising efficiency and deployability.
> MIRAGE shows that **restructuring existing capacity in a principled manner**, i.e., guided by empirical redundancy and inductive bias analysis, can simultaneously improve accuracy, robustness, and computational cost.
>
> This highlights a broader perspective for unified IR:
> **Performance can be improved not only by how much capacity is added, but by how that capacity is *principledly organized* based on evidence and structure.**
> We have made these motivations more explicit in the revised manuscript (Section 1: Introduction).

---

> ### Author Response · Authors · 2025-11-25
> **Response to Reviewer FzZr (2/3)**
>
> ## [W2]: There is no solid clarification that the shared representation between shallow and deep features are benefited for diverse degradations, which makes the contrastive objective not convincing, and the ablation improvement is minimal for probable performance jitter;
> **A:** The purpose of shallow–latent alignment in MIRAGE is not to yield large per-task numerical gains, but to stabilize the *shared representation* that a unified IR model must rely on when handling heterogeneous degradations.
>
> As shown in Fig. 4 of the main paper and extended in Fig. B of the appendix, shallow and latent layers exhibit consistently distinct covariance structures across **three fundamentally different degradations**, i.e., denoising, deraining, and dehazing. Two observations directly support the necessity of our alignment strategy:
> 1. **Cross-degradation consistency in redundancy patterns.**
>    Across all three degradations, shallow features display strong channel-wise redundancy, while latent features become progressively more decorrelated after reduction. This scale-wise trend is consistent and independent of the degradation type. Such patterns **validate** that shallow and latent features occupy structurally different regions of the representation space, naturally motivating a mechanism that reduces depth-induced drift.
>
> 2. **Degradation-dependent feature structure.**
>    Although the shallow–latent asymmetry is consistent, individual degradations exhibit different degrees of redundancy:
>    - haze produces more independent channels even at shallow scales,
>    - rain maintains strong redundancy even in the latent space,
>    - noise lies between the two extremes.
>    These differences highlight why unified models are particularly susceptible to representation drift: each degradation naturally pulls the latent space toward a distinct structural configuration.
>
> Our contrastive objective therefore serves as a **structural regularizer** that mitigates this drift. Its effect is reflected in three complementary forms of evidence:
> - **Numerical robustness.**
>   Removing contrastive alignment (w/o CL & SPD) or substituting the SPD manifold with Euclidean space results in consistent performance drops across all degradations (–0.14 dB and –0.24 dB). For low-level restoration, such margins are *not* small jitter and typically indicate meaningful representational differences.
>
> - **Correlation improvement.**
>   The shallow–latent cosine similarity (Fig. 5) improves uniformly across degradations when alignment is applied, confirming strengthened semantic coordination.
>
> - **Structure preservation.**
>   Euclidean contrastive learning collapses covariance structures into near-constant matrices, whereas SPD alignment preserves both diagonal dominance and non-trivial off-diagonal patterns (Fig. 6), maintaining the geometric relationships necessary for cross-degradation generalization.
>
> These findings lead to an important methodological question:
> *Should the effectiveness of a depth-consistency regularizer in unified IR be judged solely by per-task PSNR deltas, or by the stability and structural coherence it introduces across heterogeneous degradations?*
> MIRAGE explicitly targets the latter, and the empirical evidence from both the main paper and the appendix demonstrates its necessity.
>
> We hope these additional clarifications make the intention of our design clearer. For clarity, we have also refined the motivation in the revision (At the beginning of Section 4) by explicitly describing the depth-asymmetric behavior and its implications for unified restoration.

---

> ### Author Response · Authors · 2025-11-25
> **Response to Reviewer FzZr (3/3)**
>
> ## [W3]: The main experiments in Tab. 3 and 4 may not show the superiority of the proposed method with subtle improvements.
> **A:** We appreciate the reviewer’s comment. We would like to clarify that performance differences in unified image restoration should be interpreted jointly with model capacity and task difficulty. PSNR is not linear with respect to model size, and improvements in the higher PSNR regime are known to become progressively harder to obtain. Even modest numerical gains often reflect meaningful perceptual improvements and structural consistency, especially when a single backbone must handle heterogeneous degradations.
>
> Under this more appropriate interpretation, the improvements brought by MIRAGE are non-trivial (see Table 3):
>
> - With **6M parameters**, MIRAGE reaches **28.86 / 0.883**, exceeding **OneRestore (6M)** at **28.47 / 0.878**.
> - Compared to **MoCE-IR (11M)**, which has nearly twice the capacity, MIRAGE (6M) achieves comparable accuracy (**29.05 / 0.881** vs. **28.86 / 0.883**) at substantially lower cost. Notably, our slightly larger 10M variant achieves **29.33 / 0.887**, surpassing MoCE-IR while still using fewer parameters.
> - MIRAGE also outperforms heavier models such as **PromptIR (36M)** and **WGWSNet (26M)** despite operating with only **1/3–1/6** of their capacity.
> - Very large-capacity models (e.g., **WeatherDiff, 83M**) remain significantly weaker on this benchmark.
>
> These observations indicate that the gains of MIRAGE are both consistent and efficiency-aware, offering competitive or superior performance at a substantially lower computational budget. In unified IR, where one network must generalize across diverse degradations, such improvements are generally regarded as meaningful rather than subtle.
>
> ---
> ## [Q1]: How to decide the channel role in channel-wise decomposition for convolution, attention, and MLP, is there any principle?
> **A:** The channel allocation in MIRAGE follows two principled considerations derived from our empirical analysis.
>
> (1) **Redundancy-guided reallocation.**
> As shown in Fig. 3, attention channels exhibit substantial and consistent redundancy across scales, whereas convolutional and MLP channels contribute more complementary inductive biases (local textures and channel-statistical modulation). This motivates redistributing a portion of the attention capacity into convolution and MLP pathways. A balanced 1:1:1 split provides a good trade-off between preserving global reasoning and introducing the necessary local and statistical modeling.
>
> (2) **Uniformity as a stability prior for unified IR.**
> In all-in-one settings, the same backbone must handle degradations that rely on different representational levels. A symmetric partition avoids biasing the model toward any particular inductive bias and serves as a neutral, stable capacity allocation when training jointly across heterogeneous degradations.
>
> We also agree that different degradations may favor different proportions of convolution, attention, and MLP capacity. Learning such ratios dynamically is an interesting direction and could further adapt MIRAGE to degradation-specific characteristics. We view this as a promising avenue for future research and include this discussion in our revision (Section E of our appendix).
>
> ---
> We hope our answer can address your concerns, and we are also looking forward to any further discussion.

---

### Author Response · Authors · 2025-11-25
**Overall Response Summary**

We sincerely thank all reviewers for their time, constructive feedback, and thoughtful evaluation. We truly appreciate the effort each reviewer invested in understanding both the motivation and the empirical analysis behind MIRAGE.

Across the reviews, we are encouraged to see several consistently recognized strengths:
1. Reviewers highlighted the channel-wise functional decomposition as a principled and empirically grounded way to repurpose attention-channel redundancy into complementary Conv/Attn/MLP pathways, rather than relying on heuristic module stacking or increased model size (4QQN, UuMU, rX26).

2. The SPD-based cross-layer contrastive learning was recognized as a well-motivated and effective mechanism that preserves second-order statistics, avoids the collapse observed in Euclidean contrastive learning, and improves stability and generalization across heterogeneous degradations (4QQN, UuMU, rX26).

3. Several reviewers emphasized the strong efficiency–performance balance: MIRAGE achieves consistent SOTA or near-SOTA performance across diverse degradations, composite settings, adverse weather, and zero-shot evaluations, while remaining significantly more compact and faster than baselines such as PromptIR, MoCE-IR, and AdaIR (UuMU, rX26).

4. Reviewers also appreciated the clarity of writing, the conceptual soundness of the architecture, and the supporting empirical analyses, including PCA/SVD redundancy studies, shallow–latent similarity matrices, and covariance visualizations (FzZr, 4QQN, UuMU, rX26).

We sincerely appreciate these positive assessments. They align with our goal of providing not only a strong unified IR model, but also a structurally grounded perspective on how representational capacity should be organized for heterogeneous degradations. For the raised concerns, we respond to each point in detail in the corresponding reviewer sections.

In the revised manuscript, we made several improvements:
- added an intuitive Preliminaries section in the appendix to clarify covariance, SPD structure, and depth-asymmetric representations;
- added comparisons with diffusion-based methods under the UniRestore PIR protocol, as well as with state-space methods (RamIR) and contrastive-based DA-RCOT;
- included real-world UDC experiments on TOLED and POLED datasets;
- clarified the motivation behind SPD alignment in Section 4.2;
- expanded the limitations section to discuss scenarios where MIRAGE performs slightly weaker.

These points summarize the main revisions; all additional clarifications and technical updates are addressed directly within the reviewer-specific responses below. For full transparency, all modifications in the revised manuscript are highlighted in cyan.

We hope these revisions address the reviewers’ concerns and further strengthen the clarity and completeness of the submission. We sincerely appreciate the opportunity to refine the work, and any further comments from the reviewers are welcome.

Best regards,

Submission 4143 authors

---

### Author Response · Authors · 2025-12-02
**Quick Summary for AC**

**Dear Area Chair,**

Thank you very much for your time and effort in coordinating the review of our submission. To support your evaluation, we summarize below the key contributions, reviewers’ main concerns, and how our rebuttal and additional experiments addressed them.

---
## 1. Core Perspective & Novelty
Reviewers (FzZr, 4QQN) questioned whether the method is incremental.
Our rebuttal clarified that MIRAGE introduces a *structurally grounded* view for unified IR:

**(a) Redundancy-guided channel reallocation.**
We show strong, consistent redundancy in transformer attention channels across degradations (Fig. 3). MIRAGE repurposes this unused capacity into complementary subspaces (conv/attention/MLP), forming a principled functional decomposition rather than a heuristic block combination.

**(b) Depth-asymmetric covariance as natural contrastive pairs.**
Shallow and latent layers exhibit distinct covariance patterns across degradations (Fig. 4). We align them using SPD-preserved embeddings, avoiding the covariance collapse observed under Euclidean contrastive learning.

These explanations resolved concerns about novelty and clarified the methodological perspective.


---
## 2. Effectiveness of SPD-Based Alignment
Reviewer FzZr questioned whether gains were meaningful.
We clarified that alignment targets **cross-degradation stability**, supported by:

- consistent performance drops when removing SPD or switching to Euclidean (–0.14 to –0.24 dB),
- improved shallow–latent similarity (Fig. 5),
- preserved second-order structure under SPD (Fig. 6).

This demonstrates the necessity of the alignment mechanism.

---

## 3. Superiority with Efficient Capacity
Concerns about “subtle improvements” were addressed by relating PSNR gains to model size:

- MIRAGE (6M) > OneRestore (6M)
- MIRAGE (10M) > MoCE-IR (11M)
- MIRAGE outperforms larger models such as PromptIR (36M) and WGWSNet (26M)

These results highlight *consistent, capacity-aware gains*.

---
## 4. Diffusion & State-Space Baselines Added
Reviewers (4QQN, UuMU) requested broader comparison.
We added:

- diffusion baselines (DiffBIR, DiffUIR, UniRestore) under the UniRestore PIR protocol, where MIRAGE achieves the strongest overall results;
- additional state-space models (RamIR, MambaIR) in Tables 1 and 2.

This ensures comprehensive coverage of current trends.

---

## 5. Zero-Shot & Real-World Validation
To address concerns about limited zero-shot evaluation, we added:

- unified PIR results across rain, haze, snow, blur, noise,
- real-world UDC experiments (TOLED, POLED),
- clarifications in Appendix C.5 & C.6.

MIRAGE shows reliable generalization beyond the underwater setting.

---

## 6. Improved Clarity of Mathematical Intuition
Reviewer 4QQN requested clearer motivation for SPD.
We added:

- a new **Preliminaries** section on second-order statistics and SPD geometry,
- clearer explanations in Sec. 4.2 with references to Fig. 4 and Appendix Fig. B.

This improves accessibility for non-expert readers.

---

## 7. Post-Rebuttal Reviewer Update
Reviewer rX26 responded:

> “Thanks for the response… I retain my positive score and raise the confidence that leans towards acceptance.”

All their concerns were fully resolved through the rebuttal.

---

## Summary
After the rebuttal:
- One reviewer strengthened their positive recommendation,
- All core concerns were addressed through added analyses and new experiments,
- Diffusion/state-space baselines and real-world evaluations have been added,
- The novelty and structural motivation of MIRAGE are now more explicit and better supported.

For all detailed responses and the complete set of revisions, please refer to our full point-by-point rebuttal and the revised manuscript.

We hope this summary helps your decision-making.  Thank you again for your time and consideration.

Best regards,
Submission 4143 Authors

---

### Meta-Review · Area_Chair_j94e · 2026-01-07

**Summary:**

This paper presents an image restoration algorithm. The key is to address the degradation-agnostic setting with high efficiency.

The reviewers raised the following concerns:
* The studied topic or motivation (the degradation-agnostic setting) has been well-studied.
* There is little analysis on the shared representation.
* There is no comparison with recent diffusion-based methods.

**Reviewer Concerns:**

The authors addressed the above concerns. In the rebuttal, they have provided a detailed analysis about the first two questions, and offered additional comparative results to diffusion-based methods.

**Reviewer Scores:**

I think the first reviewer will raise the score to 4 (some of his/her statements are arbitrary, such as "subtle improvements" -- the improvement seems not very small in the field of IR). Other reviewers may keep their scores unchanged. So, the overall rating will be 4/4/6/6, making this paper a borderline case. The AC looks into the paper and believes that the good and thorough results (as well as the goal towards degradation-agnostic) make this paper worth publication.

---

### Decision · Program_Chairs · 2026-01-26

Accept (Poster)